# ONLINE FEATURE UPDATES IMPROVE ONLINE (GENERALIZED) LABEL SHIFT ADAPTATION

## ABSTRACT

This paper addresses the prevalent issue of label shift in an online setting with missing labels, where data distributions change over time and obtaining timely labels is challenging. While existing methods primarily focus on adjusting or updating the final layer of a pre-trained classifier, we delve into the untapped potential of enhancing feature representations using unlabeled data at test-time. Our novel Online Label Shift adaptation with Online Feature Updates (OLS-OFU) method harnesses self-supervised learning to refine the feature extraction process, thus improving the prediction model. Theoretical analyses confirm that OLS-OFU reduces algorithmic regret by capitalizing on self-supervised learning for feature refinement. Empirical tests on CIFAR-10 and CIFAR-10C datasets, under both online label shift and generalized label shift conditions, underscore OLS-OFU's effectiveness and robustness, especially in cases of domain shifts.

## 1 INTRODUCTION

The effectiveness of most supervised learning models relies on a key assumption that the train data and test data share the same distribution. However, this assumption rarely holds in real-world scenarios, giving rise to *distribution shift*. Previous research has primarily focused on understanding distribution shifts in offline or batch settings, where a single shift occurs between the train and test distributions. In contrast, real-world applications often involve test data arriving in an *online* fashion, and the distribution shift can continuously evolve over time. Additionally, there is another challenging issue of *missing and delayed* feedback labels, stemming from the online setup, where gathering labels for the streaming data in a timely manner becomes a challenging task.

To tackle the distribution shift problem, prior work makes further assumptions on the nature of the shift, such as label shift or covariate shift. In this paper, we focus on the common *(generalized) label shift* problem in an online fashion with missing labels. Specifically, the learner is given a fixed set of labeled training data $D_0 \sim \mathcal{P}^{\text{train}}$ in advance and trains a model $f_0$. At test-time, only a small batch of unlabelled test data $S_t \sim \mathcal{P}_t^{\text{test}}$ arrives in an online fashion ($t = 1, 2, \cdots$). For online label shift, we assume the label distribution $\mathcal{P}_t^{\text{test}}(y)$ may change over time $t$ while the conditional distribution stays the same, i.e. $\mathcal{P}_t^{\text{test}}(x|y) = \mathcal{P}^{\text{train}}(x|y)$. MRI image classifiers for concussion detection can be challenging due to label shift caused by seasonal changes in the image distribution. A classifier trained during skiing season may perform poorly when tested afterward, as the image distribution changes continuously between skiing and non-skiing seasons. The *generalized* label shift relaxes this assumption by assuming there exists a transformation $h$ of the covariate, such that the conditional distribution $\mathcal{P}_t^{\text{test}}(h(x)|y) = \mathcal{P}^{\text{train}}(h(x)|y)$ stays the same. Reiterating our example, a classifier on MRI images can be used in different clinics during training and testing, where the MRI machines might have different versions at both hardware and software levels. Then, the images may have some variations such as brightness, resolution, etc. However, a feature extractor $h$ exists, capable of mapping these variations to the same point in the latent space. The goal of the learner is to adapt to the (generalized) label shift within the non-stationary environment, continually adjusting the model's predictions in real time.

Existing online label shift adaptation algorithms (OLS) primarily adopt one of two strategies: either directly reweighting of the pretrained classifier $f_0$, or re-training only the final linear layer of $f_0$ — typically keeping the feature extractor frozen. Recent work has demonstrated that feature extractors can still be improved, even during test-time and in the absence of labels. We hypothesize that a

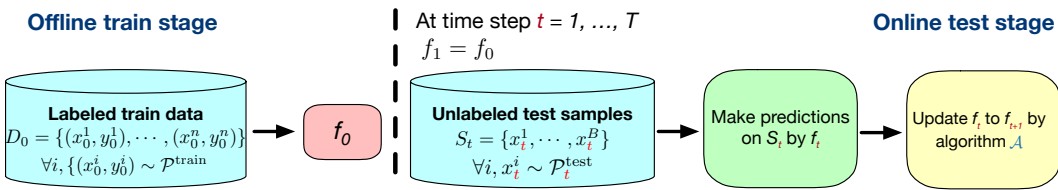

Figure 1: Overview of online distribution shift adaptation. We further assume $\mathcal{P}_t^{\text{test}}(x|y) = \mathcal{P}^{\text{train}}(x|y)$ for online label shift adaptation and assume the existence of unknown feature transformation $h$ such that $\mathcal{P}_t^{\text{test}}(h(x)|y) = \mathcal{P}^{\text{train}}(h(x)|y)$ for online generalized label shift adaptation.

similar effect can be leveraged in the (generalized) label shift setting and propose to improve the feature representation during testing. In online label shift, updating the feature extractor offers two possible advantages. First, it enables the utilization of additional unlabeled samples to enhance the sample efficiency of the feature extractor. Second, it allows the adaptation of the feature extractor to label shift. Latter is important because the optimal feature extractor is not necessarily independent of the label distribution. In particular, in generalized label shift, the feature transformation $h$ is typically unknown, and additional unlabeled test samples can ease the learning of $h$.

Building upon this insight, this paper introduces the *Online Label Shift adaptation with Online Feature Updates* (OLS-OFU) framework, aimed at enhancing feature representation learning in the context of online label shift adaptation. Specifically, each instantiation of OLS-OFU incorporates a self-supervised learning method associated with a loss function denoted as $l_{\text{ssl}}$ and an existing online label shift adaptation (OLS) algorithm. In each time step, it executes a modified version of OLS, updates the feature extractor through self-supervised learning, and subsequently refines the last linear prediction layer. OLS-OFU is easy to implement and can be seamlessly integrated with any off-the-shelf OLS algorithms. Theoretically, we demonstrate that OLS-OFU reduces the loss of the overall algorithm by leveraging self-supervised learning techniques to enhance the feature extractor, thereby improving predictions for test samples at each time step $t$. Empirical evaluations on CIFAR-10 and CIFAR-10C datasets, considering both online label shift and online generalized label shift scenarios, affirm the effectiveness of OLS-OFU. It outperforms its counterpart OLS method across various distribution shift settings and consistently across all OLS methods, underscoring its robustness. Notably, the advantages of OLS-OFU become particularly pronounced in cases of domain shifts that happened in the generalized label shift, as demonstrated in the ablation study.

## 2 PROBLEM SETTING & RELATED WORK

We start with some basic notations. Let $\Delta^{K-1}$ be the probability simplex. Let $f : \mathcal{X} \to \Delta^{K-1}$ denote a classifier. Given an input $x$ from domain $\mathcal{X}$, $f(x)$ outputs a probabilistic prediction on $K$ classes. For example, $f$ can be the output from the softmax operation after any neural network. If we reweight a model $f$ by a vector $p \in \mathbb{R}^K$, we refer to this model as $g(\cdot; f, p)$ with $g$ denotes the method of reweighting. For any two vectors $p$ and $q$, $p/q$ denotes the element-wise division.

**Online distribution shift adaptation.** The effectiveness of any machine learning model $f$ relies on a common assumption that the train data $D_0$ and test data $D_{\text{test}}$ are sampled from the same distribution, i.e., $\mathcal{P}^{\text{train}} = \mathcal{P}^{\text{test}}$. However, this assumption is often violated in practice, which leads to *distribution shift*. This can be caused by various factors, such as data collection bias and changes in the data generation process. Moreover, once a well-trained model $f_0$ is deployed in the real world, it enters the test stage, which can be a sequence of time periods or time steps. The test distribution at time step $t$, $\mathcal{P}_t^{\text{test}}$, from which test data $x_t$ is sampled, may vary over time. One example is that an MRI image classifier might be trained on MRI images during skiing season (which may have a high frequency of head concussions) but tested afterward (where the frequency of concussion is lower). The test stage can be several months long until the next classifier is trained. The MRI image distribution during the test stage may change continuously between non-skiing and skiing seasons.

As the test-time distribution changes over time, the challenge lies in how to adjust the model continuously from $f_{t-1}$ to $f_t$ in an online fashion to adapt to the current distribution $\mathcal{P}_t^{\text{test}}$. We call this problem *online distribution shift adaptation* and illustrate it in Figure 1. Given a total $T$ steps in

the online test stage, we define the average loss for any online algorithm $\mathcal{A}$ through the loss of the sequence of models $f_t$, $t \in [T]$ that are produced from $\mathcal{A}$, i.e.,

$$L(\mathcal{A}; \mathcal{P}_1^{\text{test}}, \cdots, \mathcal{P}_T^{\text{test}}) = \frac{1}{T} \sum_{t=1}^{T} \ell(f_t; \mathcal{P}_t^{\text{test}}), \tag{1}$$

where $\ell(f; \mathcal{P}) = \mathbb{E}_{(x,y) \sim \mathcal{P}} \ell_{\text{sup}}(f(x), y)$ and $\ell_{\text{sup}}$ is the loss function, for example, 0-1 loss or cross-entropy loss for classification tasks.

If we have knowledge of $\mathcal{P}_t^{\text{test}}$ or have enough samples from $\mathcal{P}_t^{\text{test}}$, then the problem reduces to offline distribution shift for each single time step, or if we have a few labeled samples, the problem can be treated as classical online learning, and we can run stochastic gradient descent at the end of every time step $t$ to meet certain theoretical guarantees. However, it is more realistic but technically highly non-trivial when we only have very few unlabeled samples, or even a single sample $x_t$. An effective algorithm under this scenario has to utilize information from all historical data (both train data $D_0$, validation data $D_0'$ and test data up to time $t$), as well as previously deployed models $f_{0,1,\cdots,t-1}$. In this paper, we consider this challenging case where at each time step $t$, only *a small batch of unlabeled samples $S_t = \{x_t^1, \cdots, x_t^B\}$* is received. We formalize the algorithm $\mathcal{A}$ as:

$$f_t := \mathcal{A}\left(\{S_1, \cdots, S_{t-1}\}, \{f_0, f_1, \cdots, f_{t-1}\}, D_0, D_0'\right) \quad \forall t \in [T]. \tag{2}$$

In contrast to the classical online learning setup, this scenario presents a significant challenge as classical online learning literature usually relies on having access to either full or partial knowledge of the loss at each time step, i.e., $\ell_{\text{sup}}(f(x_t), y_t)$. In this setting, however, only a batch of unlabeled samples is provided at each time step. This lack of both label information and loss values significantly amplifies the difficulty of accurately estimating the true loss in Equation 1.

Related work have studied the online distribution shift adaptation under various assumptions. Zhang et al. (2023) assumes the test distributions are covariate shifted from the training distribution, i.e. $\mathcal{P}_t^{\text{test}}(y|x) = \mathcal{P}_t^{\text{train}}(y|x)$ while $\mathcal{P}_t^{\text{test}}(x) \neq \mathcal{P}^{\text{train}}(x)$ may shift over time. Wu et al. (2021); Bai et al. (2022); Baby et al. (2023) study online label shift adaptation, where $\mathcal{P}_t^{\text{test}}(x|y) = \mathcal{P}^{\text{train}}(x|y)$ and $\mathcal{P}_t^{\text{test}}(y) \neq \mathcal{P}^{\text{train}}(y)$ can shift over time. In this paper, we focus on the online (generalized) label shift adaptation. Generalized label shift, first introduced by Tachet des Combes et al. (2020), assumes that there exists an unknown transformation $h$ of the feature $x$, such that the conditional distribution $\mathcal{P}(h(x)|y)$ remains invariant between the training and test time. Discussed in Section 5, our methodology applies beyond label shift scenarios, extending to distribution shift more broadly.

**Online label shift adaptation.** In online label shift, the conditional distribution $\mathcal{P}_t^{\text{test}}(x|y)$ is invariant and equivalent to $\mathcal{P}^{\text{train}}(x|y)$ for all $t \in [T]$, while the marginal distribution of the label $\mathcal{P}_t^{\text{test}}(y)$ changes over time. This assumption is most typical when the label $y$ is the causal variable and and the feature $x$ is the observation (Schölkopf et al., 2012) and the example of concussion detection from MRI images lies in the case. Most previous methods tackle this problem by a non-trivial reduction to the classical online learning problem. Given this, most of the online label shift algorithms (Wu et al., 2021; Bai et al., 2022; Baby et al., 2023) study the theoretical guarantee of the algorithm via the convergence of the regret function, either static regret or dynamic regret. In prior studies, the hypothesis class $\mathcal{F}$ of the prediction function $f$ is typically chosen in one of two ways. The first approach involves defining $\mathcal{F}$ as a family of post-hoc reweightings of $f_0$, with the parameter space comprising potential reweight vectors. Notable examples within this category include ROGD (Wu et al., 2021), FTH (Wu et al., 2021), and FLHFTL (Baby et al., 2023). The second approach defines $\mathcal{F}$ as a family of functions that share the same parameters in $f_0$ except the last linear layer, for example, UOGD (Bai et al., 2022) and ATLAS (Bai et al., 2022).

**Online generalized label shift adaptation.** In the context of MRI image classification, where head MRI images serve as the feature $x$, MRI machines in different clinics may use different versions of software or hardware. The resulting images may exhibit differences in terms of brightness, contrast, resolution, and other characteristics. In such scenarios, the conditional probability distribution $\mathcal{P}(x|y)$ is no longer invariant. However, when a feature extractor $h$ is robust enough, it can map the images into a feature space where the images from different machines have the same distributions $\mathcal{P}(h(x)|y)$ in this transformed feature space. The concept of generalized label shift, as introduced in Tachet des Combes et al. (2020), formalizes this situation by postulating the existence of an unknown function $h$, such that the conditional probability distribution $\mathcal{P}(h(x)|y)$ remains invariant. The primary challenge in this context is to find the underlying transformation $h$. Building upon this,

online generalized label shift assumes that, for every time step $t$, the test distribution $\mathcal{P}_t^{\text{test}}$ exhibits a form of generalized label shift from the training distribution $\mathcal{P}^{\text{train}}$, and this shift is governed by the same underlying transformation $h$.

**Self-supervised learning.** Inspired by the body of work in semi-supervised learning (Grandvalet & Bengio, 2004b; Lee et al., 2013; Laine & Aila, 2016; Gidaris et al., 2018; Miyato et al., 2018) and unsupervised representation learning (Chen et al., 2020a; He et al., 2019; Grill et al., 2020; He et al., 2022), self-supervised learning (SSL) techniques emerge as promising tools for enhancing feature extraction from unlabeled data, e.g. for image classification. Likewise, when dealing with online label shift adaptation, it is crucial to leverage the unlabeled test samples $S_1 \cup \cdots \cup S_{t-1}$ obtained from previous time steps. Ideally, these unlabeled samples could improve the feature representation learning and ultimately lead to better prediction for the test samples $S_t$ at a time step $t$.

## 3 METHOD

We introduce a novel online label shift adaptation algorithm that uses self-supervised learning (SSL) to improve representation learning. This approach is general and can be used with any existing OLS algorithm. We show theoretically and empirically the performance improvement in OLS algorithms and their effectiveness for the more challenging online generalized label shift adaptation problem.

### 3.1 ONLINE FEATURE UPDATES WITH SELF-SUPERVISED LEARNING

In this section, we discuss how to utilize various SSL techniques to improve the feature representation learning of existing online label shift algorithms. To illustrate the concept, we narrow our focus to three particular SSL techniques for classification tasks: rotation degree prediction (Gidaris et al., 2018; Sun et al., 2020), entropy minimization (Grandvalet & Bengio, 2004b; Wang et al., 2020) and MoCo (He et al., 2019; Chen et al., 2020b; 2021). It is important to note that this concept extends beyond these three SSL techniques, and the incorporation of more advanced SSL techniques has the potential to further elevate the performance. Specifically, rotation degree prediction involves initially rotating a given image by a specific degree from the set $\{0, 90, 180, 270\}$ and the classifier is required to determine the degree by which the image has been rotated. Entropy minimization utilizes a minimum entropy regularizer, with the motivation that unlabeled examples are mostly beneficial when classes have a small overlap. MoCo is a more advanced representation learning technique, using a query and momentum encoder to learn representations from unlabeled data by maximizing the similarity of positive pairs and minimizing the similarity of negative pairs. The details of these SSL are introduced in Appendix D.8.

Now we formally introduce *Online Label Shift adaptation with Online Feature Updates* (OLS-OFU; Algorithm 1), which requires a self-supervised learning loss $\ell_{\text{ssl}}$ and an online label shift adaptation algorithm (OLS) that either reweights the offline pretrained model $f_0$ or updates the last linear layer[1]. In the train stage, we train $f_0$ by minimizing the supervised and self-supervised loss together defined on train data. In the test stage, OLS-OFU comprises three steps at each time step $t$: (1) running the refined version of OLS, which we refer to as OLS-R, (2) updating the feature extractor, (3) re-training the last linear layer. The details of these three steps are elaborated below.

**(1) Running the Revised OLS.** First, we review common OLS methods (FLHFTL, FTH, ROGD, UOGD, and ATLAS) and identify two specific points in the algorithm where we can employ the updated prediction model $f_t''$ (with the improved feature extractor[2]) to supplant the pretrained model $f_0$, hence enhancing the original OLS algorithm. Denote $C_{f,D} \in [0,1]^{K \times K}$ the confusion matrix evaluated on dataset $D$ for the model $f$ with $C_{f,D}[i,j] = \mathbb{P}_{(x,y)\sim D}\big(\arg\max(f(x)) = j|y = i\big)$. At the outset, all OLS methods rely on an unbiased estimator $s_t$ of the label distribution $q_t$ with $q_t[y] = \mathcal{P}_t^{\text{test}}(y)$, where $s_t := \frac{1}{|S_t|} \sum_{x_t \in S_t} C_{f_0,D_0}^{-1} f_0(x_t)$. This is the first point that we can replace $f_0$ with the improved prediction model $f_t''$ to enhance the estimation of label marginal distribution. For the second point,

---

[1]As pointed out in Section 2, this is relatively general as most previous online label shift algorithms belong to these two categories.

[2]We detail how to obtain this enhanced model $f_t''$ using SSL techniques in the following paragraphs.

---

**Algorithm 1** Online label shift adaptation with online feature updates (OLS-OFU).

---

**Require:** An online label shift adaptation algorithm OLS ($\in$\{ROGD, FTH, UOGD, ATLAS, FLHFTL\}), a self-supervised learning loss $\ell_{\mathrm{ssl}}$. A pretrained model $f_0$ and initialize $f_1 = f_0$.

**for** $t = 1, \cdots, T$ **do**

    **Input at time** $t$**:** Samples $S_1 \cup \cdots \cup S_t$, models $\{f_1, \cdots, f_t\}$, train set $D_0$ and validation set $D_0'$.

    1. Run the revised version of OLS, that is, OLS-R, and get $f_{t+1}'$. (Algorithm 2 is the revised FLHFTL; See Appendix B for revisions of ROGD, FTH, UOGD, ATLAS)

    2. Update the feature extractor $\theta_t^{\mathrm{feat}}$ in $f_{t+1}'$ by

$$\theta_{t+1}^{\mathrm{feat}} := \theta_t^{\mathrm{feat}} - \eta \cdot \nabla_{\theta^{\mathrm{feat}}} \ell_{\mathrm{ssl}}(S_t; f_{t+1}').$$

    Replace the feature extractor $\theta_t^{\mathrm{feat}}$ in $f_{t+1}'$ by $\theta_{t+1}^{\mathrm{feat}}$.

    3. Within the feature extractor $\theta_{t+1}^{\mathrm{feat}}$, re-train the last linear layer from random initialization by minimizing the average loss among train data $D_0$:

$$\theta_{t+1}^{\mathrm{linear}} := \arg \min_{\theta_{\mathrm{linear}}} \sum_{(x,y) \in D_0} \ell_{\mathrm{ce}}\left( f(x|\theta_{t+1}^{\mathrm{feat}}, \theta^{\mathrm{linear}}), y \right).$$

    Calibrate the model $f(\cdot|\theta_{t+1}^{\mathrm{feat}}, \theta_{t+1}^{\mathrm{linear}})$ by temperature calibration using the validation set $D_0'$ and denote the model after calibration as $f_{t+1}''$.

    If the parameter space of OLS is a reweighting version of the prediction model (ROGD, FTH, FLHFTL), suppose the reweighting vector in $f_{t+1}'$ is $p_{t+1}$ and we define $f_{t+1} := g(\cdot; f_{t+1}'', p_{t+1})$; else (UOGD, ATLAS), we define $f_{t+1} := f_{t+1}'$.

**end for**

---

**Algorithm 2** Revised FLHFTL for online feature updates (FLHFTL-R); See the original version in Algorithm 2 in Baby et al. (2023).

---

**Require:** Online regression oracle ALG.

**for** $t = 1, \cdots, T$ **do**

    **Input at time** $t$**:** Samples $S_1 \cup \cdots \cup S_t$, models $\{f_1, \cdots, f_t\}$, and intermediate model $\{f_1'', \cdots, f_t''\}$ from step 3 in Algorithm 1, the validation set $D_0'$, the train label marginal $q_0 := \mathcal{P}^{\mathrm{train}}(y)$.

    1. Compute the unbiased estimator for label marginal distribution:

      $s_t = \frac{1}{|S_t|} \sum_{x_t \in S_t} C_{f_t'', D_0'}^{-1} f_t''(x_t)$        $\triangleright$ In the original FLHFTL, it is $f_0$ rather than $f_t''$.

    2. Compute the online estimator $\tilde{q}_{t+1} := \mathrm{ALG}(s_1, \cdots, s_t)$

    3. Let $f_{t+1}'$ be a reweighting version of $f_t''$ by the weight

      $\left( \frac{\tilde{q}_{t+1}[k]}{q_0[k]} : k = 1, \cdots K \right)$        $\triangleright$ In the original FLHFTL, it is $f_0$ rather than $f_t''$.

    **Output at time** $t$**:** $f_{t+1}'$.

**end for**

---

- FLHFTL and FTH subsequently employ a vector $\tilde{q}_t$[3] to reweight the initial pretrained model $f_0$. Now, instead of reweighting the original pretrained model $f_0$, the algorithm reweights the improved model $f_t''$, utilizing its enhanced predictive performance.
- ROGD, UOGD, and ATLAS initially update the model through an unbiased gradient estimator within a hypothesis space that either includes a linear model after a fixed feature extractor in $f_0$ or a reweighted version of $f_0$. Now, we have the flexibility to shift to a comparable hypothesis space, replacing $f_0$ with the improved model $f_t''$, and continue applying updates using the same unbiased gradient estimator.

Algorithm 2 provides a formal illustration of the revision specific to FLHFTL. Further details regarding the revisions for ROGD, FTH, UOGD, and ATLAS can be found in Appendix B. We use $f_{t+1}'$ to denote the model after running the revised OLS.

---

[3]The reweighting factor $\tilde{q}_t$ is a function of unbiased estimators $s_1, \cdots, s_t$ in FLHFTL and FTH.

**(2) Updating the Feature Extractor.** We now introduce how to utilize an SSL loss $\ell_{\mathrm{ssl}}$ to update the feature extractor for any incoming unlabeled test data batch $S_t$ at timestep $t$. Specifically, let $\theta_t^{\mathrm{feat}}$ denote the parameters of the feature extractor in $f'_{t+1}$. The update of $\theta_{t+1}^{\mathrm{feat}}$ at time $t$ is given by:

$$\theta_{t+1}^{\mathrm{feat}} := \theta_t^{\mathrm{feat}} - \eta \cdot \nabla_{\theta^{\mathrm{feat}}} \ell_{\mathrm{ssl}}(S_t; f'_{t+1}).$$

We replace the feature extractor in $f'_{t+1}$ by $\theta_{t+1}^{\mathrm{feat}}$.

**(3) Re-training Last Linear Layer.** Given the updated feature extractor $\theta_{t+1}^{\mathrm{feat}}$, it is necessary to re-train the last linear layer $\theta_{t+1}^{\mathrm{linear}}$ to adapt to the new feature extractor. We start the re-raining from random initialization, while keeping the feature extractor frozen. The train objective of $\theta_{t+1}^{\mathrm{linear}}$ is to minimize the average cross-entropy loss under train data $D_0$. We denote the model with the frozen feature extractor $\theta_{t+1}^{\mathrm{feat}}$ as $f(\cdot|\theta_{t+1}^{\mathrm{feat}}, \theta^{\mathrm{linear}})$. The objective for re-training last linear layer can be written as follows:

$$\theta_{t+1}^{\mathrm{linear}} := \arg\min_{\theta_{\mathrm{linear}}} \sum_{(x,y)\in D_0} \ell_{\mathrm{ce}}\left( f(x|\theta_{t+1}^{\mathrm{feat}}, \theta^{\mathrm{linear}}), y \right).$$

We calibrate the model $f(\cdot|\theta_{t+1}^{\mathrm{feat}}, \theta_{t+1}^{\mathrm{linear}})$ by temperature calibration (Guo et al., 2017) using the validation set $D'_0$ and denote the model after calibration as $f''_{t+1}$. This re-training step is needed to ensure the model is a calibrated model of estimating $\mathcal{P}^{\mathrm{train}}(y|x)$ for any given input $x$.

In the end, we are going to define $f_{t+1}$ for the next time step. If the parameter space of the original OLS is a reweighting version of the prediction model (ROGD, FTH, FLHFTL), suppose the reweighting vector in $f'_{t+1}$ is $p_{t+1}$ and we define $f_{t+1} := g(\cdot; f''_{t+1}, p_{t+1})$; else (UOGD, ATLAS), we define $f_{t+1} := f'_{t+1}$.

## 3.2 PERFORMANCE GUARANTEE

The original Online Label Shift (OLS) methods exhibit theoretical guarantees in terms of regret convergence for online label shift setting, where $\mathcal{P}_t^{\mathrm{test}}(x|y) = \mathcal{P}^{\mathrm{train}}(x|y)$ is invariant. With the incorporation of the additional online feature update step, OLS-OFU demonstrates analogous theoretical results. By comparing the theoretical results between OLS and OLS-OFU, we can gain insights into potential enhancements from OLS to OLS-OFU. Due to limited space, we provide the theoretical results pertaining to FLHFTL-OFU here, and present the theoretical results for ROGD-OFU, FTH-OFU, UOGD-OFU, and ATLAS-OFU in Appendix C.

**Theorem 1** *[Regret convergence for FLHFTL-OFU] Suppose we choose the OLS subroutine in Algorithm 2 to be FLH-FTL from Baby et al. (2023). Let $f_t^{\mathrm{flhftl-ofu}}$ be the output at time step $t-1$ from Algorithm 1, that is $g(\cdot; f''_t, \tilde{q}_t/q_0)$. Let $\sigma$ be the smallest among the the minimum singular values of invertible confusion matrices $\{C_{f''_1, D'_0}, \cdots C_{f''_T, D'_0}\}$. Then under Assumptions 1 and 2 in Baby et al. (2023), FLHFTL-OFU has the guarantee for online label shift below:*

$$\mathbb{E}\left[ \frac{1}{T} \sum_{t=1}^{T} \ell(f_t^{\mathrm{flhftl-ofu}}; \mathcal{P}_t^{\mathrm{test}}) - \frac{1}{T} \sum_{t=1}^{T} \ell(g(\cdot; f''_t, q_t/q_0); \mathcal{P}_t^{\mathrm{test}}) \right] \leq O\left( \frac{K^{1/6} V_T^{1/3}}{\sigma^{2/3} T^{1/3}} + \frac{K}{\sigma\sqrt{T}} \right), \tag{3}$$

*where $V_T := \sum_{t=1}^{T} \|q_t - q_{t-1}\|_1$, $K$ is the number of classes, and the expectation is taken w.r.t. randomness in the revealed co-variates. This result is attained without prior knowledge of $V_T$.*

To ease comparison, we state the theorem for the original OLS algorithm FLHFTL.

**Theorem 2** *[Regret convergence for FLHFTL (Baby et al., 2023)] Under Assumptions 1 and 2 in Baby et al. (2023), FLHFTL has the guarantee for online label shift below:*

$$\mathbb{E}\left[ \frac{1}{T} \sum_{t=1}^{T} \ell(f_t^{\mathrm{flhftl}}; \mathcal{P}_t^{\mathrm{test}}) \right] - \frac{1}{T} \sum_{t=1}^{T} \ell(g(\cdot; f_0, q_t/q_0); \mathcal{P}_t^{\mathrm{test}}) \leq O\left( \frac{K^{1/6} V_T^{1/3}}{\sigma^{2/3} T^{1/3}} + \frac{K}{\sigma\sqrt{T}} \right), \tag{4}$$

*where $V_T := \sum_{t=1}^{T} \|q_t - q_{t-1}\|_1$, $\sigma$ denotes the minimum singular value of invertible confusion matrices $C_{f_0, D'_0}$, $K$ is the number of classes, and the expectation is taken with respect to randomness in the revealed co-variates. Further, this result is attained without prior knowledge of $V_T$.*

Recall that the objective function for the online label shift problem is defined as the average loss in Equation 1. Both theorems establish the convergence of this average loss. In the event that $f_t''$ ($t \in [T]$) from the online feature updates yield improvements:

$$\mathbb{E}\left[\frac{1}{T}\sum_{t=1}^{T}\ell(g(\cdot; f_t'', q_t/q_0); \mathcal{P}_t^{\text{test}})\right] < \frac{1}{T}\sum_{t=1}^{T}\ell(g(\cdot; f_0, q_t/q_0); \mathcal{P}_t^{\text{test}}), \qquad (5)$$

then it guarantees that the loss of FLHFTL-OFU will converge to a smaller value, resulting in enhanced performance compared to FLHFTL. We substantiate this improvement through empirical evaluation in Section 4. For other OLS algorithms such as ROGD-OFU, FTH-OFU, UOGD-OFU, and ATLAS-OFU, a similar analysis can be derived and we present them in Appendix C.

### 3.3 ONLINE FEATURE UPDATES IMPROVE ONLINE GENERALIZED LABEL SHIFT ADAPTATION

When we have knowledge of the feature map $h$ under generalized label shift, the problem simplifies to the classical online label shift scenario. However, the more challenging situation arises when $h$ remains unknown. Due to the violation of the label shift assumption, standard OLS algorithms can perform arbitrarily bad. Fortunately, existing research in test-time training (TTT) (Sun et al., 2020; Wang et al., 2020; Liu et al., 2021; Niu et al., 2022) demonstrates that feature updates driven by SSL align the source and target domains in feature space. When the source and target domains achieve perfect alignment, such feature extractor effectively serves as the feature map $h$ as assumed in generalized label shift. Therefore, the sequence of feature extractors in $f_1, \cdots, f_T$ generated by Algorithm 1 progressively approximates the underlying $h$. This suggests that, compared to the original OLS, OLS with online feature updates (Algorithm 1) experiences a milder violation of the label shift assumption within the feature space and is actually expected to have better performance in the setting of online generalized label shift. Indeed, as demonstrated in Section 4, we can observe significant improvements in OLS-OFU compared to standard OLS for online generalized label shift.

## 4 EXPERIMENT

In this section, we empirically evaluate how OLS-OFU improves the original OLS methods on both online label shift and online generalized label shift[4]. The experiment is performed on CIFAR-10 (Krizhevsky et al., 2009) and CIFAR-10C (Hendrycks & Dietterich, 2019), where we vary the shift processes and SSL techniques to evaluate the efficacy of the method.

### 4.1 EXPERIMENT SET-UP

**Dataset and shift process set-up.** For online label shift, we evaluate the efficacy of our algorithm on CIFAR-10, which has 10 categories of images. We split the original train set of CIFAR-10 into the offline train and validation sets, which have 40,000 and 10,000 images respectively. At the online test stage, the unlabeled batches are sampled from the test set of CIFAR-10. For online generalized label shift, the offline train and validation sets are the same CIFAR-10 images, but the test unlabeled batches are sampled from CIFAR-10C. CIFAR-10C is the benchmark that has the same objects in CIFAR-10 but with multiple types of corruption. We experiment with three types of corruptions (i.e., domain shifts): Gaussian noise, Fog and Pixelate. Besides CIFAR-10 and CIFAR-10C, we also experiment with three additional datasets for the setting of online label shift and present the results in Appendix D.2 due to the space limits. See more details of dataset set-up in Appendix D.1.

We follow Bai et al. (2022) and Baby et al. (2023) to simulate the online label distribution shift in two online shift patterns: Sinusoidal shift and Bernoulli shift. Given two label distribution vectors $q$ and $q'$, the label marginal distributions at time $t$ is $q_t := \alpha_t q + (1 - \alpha_t)q'$. In Sinusoidal shift, $\alpha_t = \sin\frac{i\pi}{L}$ (periodic length $L = \sqrt{T}$, $i = t \mod L$) while in Bernoulli shift, $\alpha_t$ is a random bit (either 0 or 1), where the bit switches $\alpha_t = \alpha_t - 1$ if the coinflip probability exceeds $p = \frac{1}{\sqrt{T}}$. The $q$ and $q'$ are $\frac{1}{K}(1, \cdots, 1)$ and $(1, 0, \cdots, 0)$ in the experiment. To sample the batch test data at time $t$, we first sample a batch of labels (not revealed to the learner) according to $q_t$. Then given each label

---

[4]Code released at https://anonymous.4open.science/r/online_label_shift_with_online_feature_updates-3A1F/.

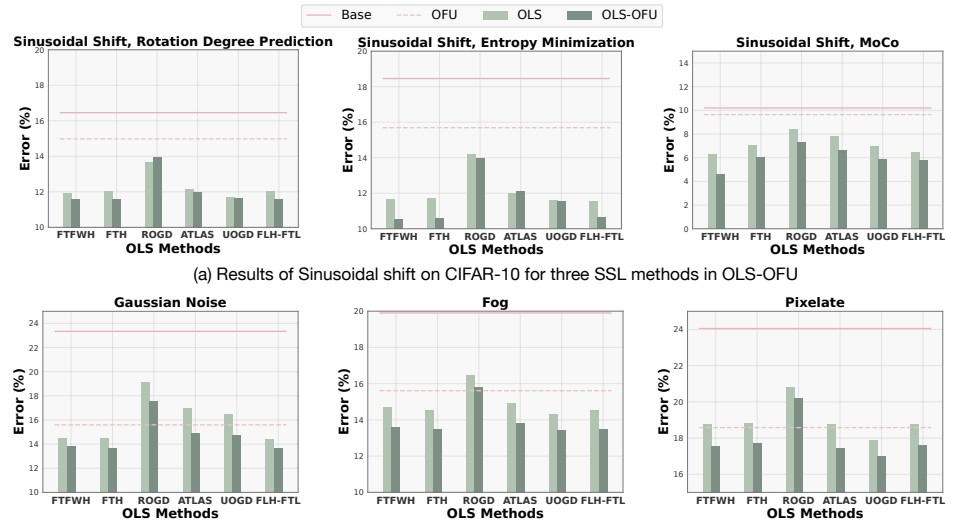

(a) Results of Sinusoidal shift on CIFAR-10 for three SSL methods in OLS-OFU

(b) Results of Sinunoidal shift on three types of corruptions in CIFAR-10C. SSL method in OLS-OFU is rotation degree prediction.

Figure 2: Comparison of OLS-OFU and OLS in CIFAR-10 and CIFAR-10C.

we can sample an image from the test set, and collect this batch of images without labels as $S_t$. We experiment with $T = 1000$ and batch size $B = 10$ at each time step, following Baby et al. (2023).

**Evaluation metric.** We report the average error $\frac{1}{TB} \sum_{t=1}^{T} \sum_{x_t \in S_t} \mathbb{1}(f_t(x_t) \neq y_t)$, where $y_t \sim \mathcal{P}_t^{\text{test}}(y|x_t)$, to approximate $\frac{1}{T} \sum_{t=1}^{T} \ell(f_t; \mathcal{P}_t^{\text{test}})$ for the evaluation efficiency. This approximation is valid for large $T$ due to its exponential concentration rate by the Azuma–Hoeffding inequality.

**Online algorithms set-up.** We perform an extensive evaluation of 6 OLS algorithms in the literature: FTFWH, FTH, ROGD, UOGD, ATLAS, and FLH-FTL. FTFWH is an empirical OLS proposed in Wu et al. (2021). We further report the performance of OLS-OFU (Algorithm 1) on top of each OLS. OLS-OFU is implemented with 3 common SSL methods: rotation degree prediction, entropy minimization, and MoCo. Additionally, we report two baseline scores. The first, denoted as Base, uses the fixed pretrained model $f_0$ to predict the labels at all test time steps. The second is online feature updates (OFU) only, where at time step $t$ we only update the features (Step 2 in Algorithm 1) without utilizing OLS algorithms.

## 4.2 RESULTS

**How does OLS-OFU perform compared with original OLS methods?** Figure 2(a) compares the performance of various OLS-OFU algorithms with their respective OLS counterparts, under classical online label shift. The experiment is performed on CIFAR10 with Sinusoidal shift. Appendix D.3 shows similar conclusions for the Bernoulli shift. First of all, both OLS and OLS-OFU show significantly better performance than Base and OFU. This demonstrates the inherent advantages of OLS methods in effectively addressing the online label shift problem. Additionally, it's worth noting that OLS-OFU, when integrated with three distinct SSL methods, consistently outperforms OLS across all six OLS methods. This underscores the importance of improving feature representation learning within label shift adaptation in OLS-OFU.

Furthermore, we report the performance of OLS-OFU in the context of online generalized label shift. In this scenario, the test images exhibit three types of domain shifts in CIFAR-10C — Gaussian noise, Fog, and Pixelation—with mild severity. We present the result under Sinusoidal shift with the SSL method as rotation degree prediction. Results for other SSL methods and online shift patterns show a similar pattern in Appendix D.4. As shown in Figure 2(b), OFU has huge improvements from Base, showing the necessity of updating feature extractors to facilitate the learning of $h$. Moreover, some OLS methods perform worse than OFU. This is expected as the underlying assumption of label shift no longer holds in this generalized label shift setting. Lastly, it is worth highlighting that OLS-OFU consistently outperforms the original OLS methods by a larger gap than the one occurring at the classic label shift setting. We hypothesize that the SSL methods within OLS-OFU aid the

feature extractor in learning the unknown mapping $h$ within the generalized label shift assumption. Intuitively, the label shift assumption in the feature space of $\theta_t^{\text{feat}}$ is violated lighter than the one in the feature space of $\theta_0^{\text{feat}}$. Given the relatively mild violation, the OLS module within OLS-OFU continues to help adapt the online label shift, explaining OLS-OFU's advantage over OFU.

**How does OLS-OFU perform under high severity of domain shift in the setting of generalized online label shift?** In Figure 3, it is clear that as the domain shift severity increases, OLS-OFU significantly enlarges the gap compared with OLS. However, it might be worth pointing out that neither OLS nor OLS-OFU are better than OFU. This is because the label shift assumption is violated so severely — even OFU cannot reduce the violation under an acceptable level, especially when the OLS module exists in OLS-OFU, as the adaptation to distribution shift is far off. For results involving higher levels of severity, different corruption types, and SSL techniques, please refer to Appendix D.5.

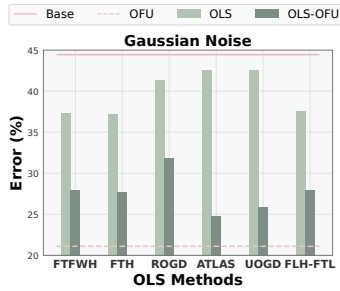

Figure 3: Results on CIFAR-10C for a high level of domain shift severity.

**Does Equation 5 hold empirically?** In Section 3.2, we argued that when the inequality in Equation 5 holds, the loss of FLHFTL-OFU exhibits a tighter upper bound compared to FL-HFTL. Figure 4 presents the RHS (corresponds to OLS) and LHS (corresponds to OLS-OFU with SSL loss as rotation degree prediction) of Equation 5. We perform the study over cross eight different settings, varying types of domain shift and online shift pattern, which empirically validates that OLS-OFU yields improvements on the *baseline* of the regret as shown in Equation 5. Appendix D.6 validates this inequality for other SSL techniques.

**Does the order of prediction and update matter?** In the default online distribution shift framework (Figure 1), model updates occur after making predictions for samples at timestep $t$. This raises the question of whether the model should be updated before making predictions. We conducted empirical evaluations for both the "predict first" and "update first" approaches and found no compelling evidence to favor one over the other (additional results in Appendix D.7). However, it's noteworthy that within the "predict first" framework, OLS and OLS-OFU benefit from robust theoretical guarantees, hence we recommend the "predict first" approach in practical applications.

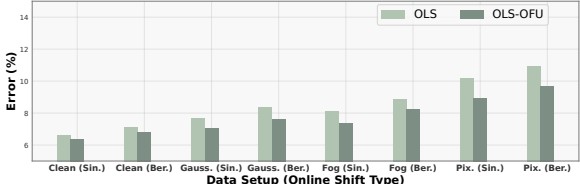

Figure 4: Empirical examination for the holdness of Equation 5. Clean denotes the experiment on CIFAR-10. Gaussion, Fog and Pixelate denote various domain shifts in CIFAR-10C. They are paired with two online shift patterns: Sinusoidal and Bernoulli.

## 5    CONCLUSION

We focus on online (generalized) label shift adaptation and introduce a novel algorithm *Online Label Shift adaptation with Online Feature Updates* (OLS-OFU), which harnesses the power of self-supervised learning to enhance feature representations dynamically during testing, leading to improved predictive models and better test time performance. Our theoretical analyses show that OLS-OFU successfully achieves the improved theoretical guarantees under online label shift. We also validate the performance of OLS-OFU on both online label shift and generalized label shift scenarios, demonstrating OLS-OFU's superiority over prior online label shift algorithms. This underscores its efficacy and robustness, particularly when confronted with domain shifts.

**Discussion and future work.** In this paper, we have shown how online feature updates help with online (generalized) label shift adaptation. One promising direction is to extend this idea to online covariate shift — the algorithm in Zhang et al. (2023) freezes the feature extractor and only updates the linear layer. Another possible extension is to consider a more realistic domain shift within the generalized label shift setting — domain shift types may vary over time or they can be more challenging, such as shifting from cartoon images to realistic images. More advanced SSL techniques in the deep learning literature might be needed to handle these more intricate domain shifts.

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

## A  FURTHER RELATED WORK

**Offline distribution shift and domain shift.** Offline label shift and covariate shift have been studied for many years. Some early work (Saerens et al., 2002; Lin et al., 2002) assumes the knowledge of how the distribution is shifted. Later work (Shimodaira, 2000; Zadrozny, 2004; Huang et al., 2006; Gretton et al., 2009; Lipton et al., 2018; Alexandari et al., 2020; Azizzadenesheli et al., 2019; Garg et al., 2020) relaxes this assumption and estimates this knowledge from unlabeled test data.

**Online distribution shift with provable guarantees.** There has been several work modeling online distribution shift as the classic online learning problem (Wu et al., 2021; Bai et al., 2022; Baby et al., 2023; Zhang et al., 2023), which leverage the classical online learning algorithms (Shalev-Shwartz, 2012; Besbes et al., 2015; Baby & Wang, 2022) to bound the static or dynamic regret. However, none of them updates the feature extractor in a deep learning model but only the last linear layer or the post-hoc linear reweighting vectors. Our proposed method OLS-OFU utilizes the deep learning SSL to improve the feature extractor, which brings better performance.

**Domain shift adaptation within online streaming data.** When we consider the most authentic online learning setup where the learner only receives the unlabeled samples, the most representative idea is test-time training (Sun et al., 2020; Wang et al., 2020; Liu et al., 2021; Niu et al., 2022), which utilizes a (deep learning) self-supervised loss to online update the model. However, it focuses on how to adapt to a *fixed* domain shifted distribution from online streaming data and is not designed for how to adapt to continuous distribution changes during the test stage, while our algorithm concentrates the later problem. Besides test-time training, Hoffman et al. (2014) and Mullapudi et al. (2019) study the online domain shift for specific visual applications.

---

**Algorithm 3** Revised ROGD for online feature updates ROGD-R. See the original version in Equation 7 and Equation 8 in Wu et al. (2021).

---

**Require:** Learning rate $\eta$.
**for** $t = 1, \cdots, T$ **do**

  **Input at time** $t$: Samples $S_1 \cup \cdots \cup S_t$, models $\{f_1, \cdots, f_t\}$, and intermediate model $\{f_1'', \cdots, f_t''\}$ from step 3 in Algorithm 1, the validation set $D_0'$, the training label marginal $q_0 := \mathcal{P}^{\text{train}}(y)$.

  1. Compute the unbiased estimator for label marginal distribution:
     $s_t = \frac{1}{|S_t|} \sum_{x_t \in S_t} C_{f_t'', D_0'}^{-1} f_t''(x_t).$  $\quad\quad$ ▷ In the original ROGD, it is $f_0$ rather than $f_t''$.

  2. Grab the weight $p_t$ from $f_t$.
  3. Update $p_{t+1} := \text{Proj}_{\Delta^{K-1}} \left[ p_t - \eta \cdot J_p(p_t)^\top s_t \right]$,
     where $J_{p,f_t''}(p_t) = \frac{\partial}{\partial p}(1 - \text{diag}(C_{f_t'', D_0, p}))|_{p=p_t}$, and let $f_{t+1}$ be a reweighting version of
     $f_t''$ by the weight $\left( \frac{p_{t+1}[k]}{q_0[k]} : k = 1, \cdots K \right)$  $\quad$ ▷ In the original ROGD, it is $f_0$ rather than $f_t''$.
  **Output at time** $t$: $f_{t+1}'$.
**end for**

---

**Algorithm 4** Revised FTH for online feature updates (FTH-R). See the original version in Equation 9 in Wu et al. (2021).

---

**for** $t = 1, \cdots, T$ **do**

  **Input at time** $t$: Samples $S_1 \cup \cdots \cup S_t$, models $\{f_1, \cdots, f_t\}$, and intermediate model $\{f_1'', \cdots, f_t''\}$ from step 3 in Algorithm 1, the validation set $D_0'$, the train label marginal $q_0 := \mathcal{P}^{\text{train}}(y)$.

  1. Compute the unbiased estimator for label marginal distribution:
     $s_t = \frac{1}{|S_t|} \sum_{x_t \in S_t} C_{f_t, D_0'}^{-1} f_t''(x_t).$  $\quad\quad$ ▷ In the original FTL, it is $f_0$ rather than $f_t''$.
  2. Compute $p_{t+1} = \frac{1}{t} \sum_{\tau=1}^{t} s_\tau$
  3. Let $f_{t+1}$ be a reweighting version of $f_t''$ by
     the weight $\left( \frac{p_{t+1}[k]}{q_0[k]} : k = 1, \cdots K \right)$  $\quad$ ▷ In the original FTL, it is $f_0$ rather than $f_t''$.
  **Output at time** $t$: $f_{t+1}'$.
**end for**

---

---

**Algorithm 5** Revised UOGD for online feature updates (UOGD-R). See the original version in Equation 9 in Bai et al. (2022).

---

**Require:** The learning rate $\eta$.
**for** $t = 1, \cdots, T$ **do**
    **Input at time** $t$**:** Samples $S_1 \cup \cdots \cup S_t$, models $\{f_1, \cdots, f_t\}$, and intermediate model $\{f_1'', \cdots, f_t''\}$ from step 3 in Algorithm 1, the validation set $D_0'$, the train label marginal $q_0 := \mathcal{P}^{\text{train}}(y)$.
    1. Compute the unbiased estimator for label marginal distribution:
    $s_t = \frac{1}{|S_t|} \sum_{x_t \in S_t} C_{f_t'', D_0'}^{-1} f_t''(x_t).$          $\triangleright$ In the original UOGD, it is $f_0$ rather than $f_t''$.

    2. Grab the weight $w_t$ from the last linear layer of $f_t$.
    3. Update $w_{t+1} := w_t - \eta \cdot \frac{\partial}{\partial w} J_w(w_t)^\top s_t$, where $J_w(w_t) = \frac{\partial}{\partial w}(\hat{R}_t^1(w), \cdots, \hat{R}_t^K(w))|_{w=w_t}$,
    $\hat{R}_t^k(w) = \frac{1}{|D_0^k|} \sum_{(x,y) \in D_0^k} \ell_{\text{ce}}(f(x|\theta_t^{\text{feat}}, \theta^{\text{linear}} = w), y)$, $D_0^k$ denotes the set of data with
    label $k$ in $D_0$.          $\triangleright$ In the original UOGD, it is $\theta_0^{\text{feat}}$ rather than $\theta_t^{\text{feat}}$.
    4. Let $f_{t+1}$ be $f(\cdot|\theta_t^{\text{feat}}, w_{t+1})$.
    **Output at time** $t$**:** $f_{t+1}'$.
**end for**

---

---

**Algorithm 6** Revised ATLAS for online feature updates (ATLAS-R). See the original version in Equation 9 in Bai et al. (2022).

---

**Require:** The learning rate pool $\mathcal{H}$ with size N; Meta learning rate $\varepsilon$; $\forall i \in [N]$, $p_{1,i} = 1/N$ and $w_{1,i} = \theta_{f_0}^{\text{linear}}$.
**for** $t = 1, \cdots, T$ **do**
    **Input at time** $t$**:** Samples $S_1 \cup \cdots \cup S_t$, models $\{f_1, \cdots, f_t\}$, and intermediate model $\{f_1'', \cdots, f_t''\}$ from step 3 in Algorithm 1, the validation set $D_0'$, the train label marginal $q_0 := \mathcal{P}^{\text{train}}(y)$.
    1. Compute the unbiased estimator for label marginal distribution:
    $s_t = \frac{1}{|S_t|} \sum_{x_t \in S_t} C_{f_t, D_0'}^{-1} f_t''(x_t).$          $\triangleright$ In the original ATLAS, it is $f_0$ rather than $f_t''$.
    **for** $i \in [N]$ **do**
        2. Update $w_{t+1,i} := w_{t,i} - \eta_i \cdot \frac{\partial}{\partial w} J_w(w_{t,i})^\top s_t$, where
        $J_w(w_{t,i}) = \frac{\partial}{\partial w}(\hat{R}_t^1(w), \cdots, \hat{R}_t^K(w))|_{w=w_{t,i}}$,
        $\hat{R}_t^k(w) = \frac{1}{|D_0^k|} \sum_{(x,y) \in D_0^k} \ell_{\text{ce}}(f(x|\theta_t^{\text{feat}}, w), y)$, $D_0^k$ denotes the set of data
        with label $k$ in $D_0$.          $\triangleright$ In the original ATLAS, it is $\theta_0^{\text{feat}}$ rather than $\theta_t^{\text{feat}}$.
    **end for**
    3. Update weight $p_{t+1}$ according to $p_{p_{t,i}} \propto \exp(-\varepsilon \sum_{\tau=1}^{t-1} \hat{R}_\tau(\mathbf{w}_{\tau,i}))$
    3. Compute $w_{t+1} = \sum_{i=1}^N p_{t+1,i} w_{t+1,i}$. Let $f_{t+1}$ be $f(\cdot|\theta_t^{\text{feat}}, w_{t+1})$.
    **Output at time** $t$**:** $f_{t+1}'$.
**end for**

---

# B    THE REVISION FOR PREVIOUS ONLINE LABEL SHIFT ADAPTATION ALGORITHMS

The revised algorithms to be used in the main algorithm OLS-OFU (Algorithm 1) are FTH-R (Algorithm 4), UOGD-R (Algorithm 5), ROGD-R (Algorithm 3), ATLAS-R (Algorithm 6).

# C    THEOREMS FOR OLS AND PROOFS

In this section, we present the theoretical results of FLHFTL-OFU, ROGD-OFU, FTH-OFU, UOGD-OFU, ATLAS-OFU and their proofs. The proofs are mostly the same as the proofs for the original algorithms with small adjustments. As our results are not straight corollaries for the original theorems, we write the full proofs here for the completeness.

## C.1 THEOREM FOR FLHFTL-OFU

Before proving Theorem 1 (in Section 3.2 ) we recall the assumption from Baby et al. (2023) for convenience. We refer the reader to Baby et al. (2023) for justifications and further details of the assumptions.

**Assumption 1** *Assume access to the true label marginals $q_0 \in \Delta_K$ of the offline train data and the true confusion matrix $C \in \mathbb{R}^{K \times K}$. Further the minimum singular value $\sigma_{min}(C) = \Omega(1)$ is bounded away from zero.*

**Assumption 2 (Lipschitzness of loss functions)** *Let $\mathcal{D}$ be a compact and convex domain. Let $r_t$ be any probabilistic classifier. Assume that $L_t(p) := E\left[\ell(g(\cdot; r_t, p/q_0)|x_t]\right.$ is $G$ Lipschitz with $p \in \mathcal{D} \subseteq \Delta_K$, i.e, $L_t(p_1) - L_t(p_2) \le G\|p_1 - p_2\|_2$ for any $p_1, p_2 \in \mathcal{D}$. The constant $G$ need not be known ahead of time.*

**Theorem 1** *[Regret convergence for FLHFTL-OFU] Suppose we choose the OLS subroutine in Algorithm 2 to be FLH-FTL from Baby et al. (2023). Let $f_t^{\text{flhftl}-\text{ofu}}$ be the output at time step $t-1$ from Algorithm 1, that is $g(\cdot; f_t'', \tilde{q}_t/q_0)$. Let $\sigma$ be the smallest among the the minimum singular values of invertible confusion matrices $\{C_{f_1'', D_0'}, \cdots C_{f_T'', D_0'}\}$. Then under Assumptions 1 and 2 in Baby et al. (2023), FLHFTL-OFU has the guarantee for online label shift below:*

$$\mathbb{E}\left[\frac{1}{T}\sum_{t=1}^{T}\ell(f_t^{\text{flhftl}-\text{ofu}}; \mathcal{P}_t^{\text{test}}) - \frac{1}{T}\sum_{t=1}^{T}\ell(g(\cdot; f_t'', q_t/q_0); \mathcal{P}_t^{\text{test}})\right] \le O\left(\frac{K^{1/6}V_T^{1/3}}{\sigma^{2/3}T^{1/3}} + \frac{K}{\sigma\sqrt{T}}\right),$$
(3)

*where $V_T := \sum_{t=1}^{T}\|q_t - q_{t-1}\|_1$, $K$ is the number of classes, and the expectation is taken w.r.t. randomness in the revealed co-variates. This result is attained without prior knowledge of $V_T$.*

**Proof:**

The algorithm in Baby et al. (2023) requires that the estimate $s_t$ in Line 1 of Algorithm 2 is unbiased estimate of the label marginal $q_t$. Since $f_t''$ in Algorithm 2 is independent of the sample $S_t$, and since we are working under the standard label shift assumption, due to Lipton et al. (2018) we have that $C_{f_t'', D_0'}^{-1} \cdot \frac{1}{|S_t|}\sum_{x_t \in S_t} f_t''(x_t)$ forms an unbiased estimate of $E_{x \sim \mathcal{P}_t^{\text{test}}}[f_t''(x)]$. Further, from Lipton et al. (2018), the reciprocal of standard deviation of this estimate is bounded below by minimum of the singular values of confusion matrices $\{C_{f_1'', D_0'}, \cdots C_{f_T'', D_0'}\}$.

Let $\tilde{q}_t$ be the estimate of the label marginal maintained by FLHFTL. By Lipschitzness, we have that

$$E[\ell(f_t^{\text{flhftl}-\text{ofu}}; \mathcal{P}_t^{\text{test}}) - \ell(g(\cdot; f_t'', p/q_0)] = E[L_t(\tilde{q}_t)] - E[L_t(q_t)] \tag{6}$$
$$\le G \cdot E[\|\tilde{q}_t - q_t\|_2], \tag{7}$$

where the last line is via Assumption 2. Rest of the proof is identical to that of Baby et al. (2023). We reproduce it below for completeness.

$$\sum_{t=1}^{T} E[\ell(f_t^{\text{flhftl}-\text{ofu}}; \mathcal{P}_t^{\text{test}}) - \ell(g(\cdot; f_t'', p/q_0)] \le \sum_{t=1}^{T} G \cdot E[\|\tilde{q}_t - q_t\|_2] \tag{8}$$

$$\le \sum_{t=1}^{T} G\sqrt{E\|\tilde{q}_t - q_t\|_2^2} \tag{9}$$

$$\le G\sqrt{T\sum_{t=1}^{T} E[\|\tilde{q}_t - q_t\|_2^2]} \tag{10}$$

$$= \tilde{O}\left(K^{1/6}T^{2/3}V_T^{1/3}(1/\sigma_{min}^{2/3}(C)) + \sqrt{KT}/\sigma_{min}(C)\right), \tag{11}$$

where the second line is due to Jensen's inequality, third line by Cauchy-Schwartz and last line by Proposition 16 in Baby et al. (2023). This finishes the proof.

**Proof of Theorem 2**: The proof is similar to the arguments in the proof of Theorem 1. The only point of deviation is that we choose $r_t = f_0$ instead of $f_t''$ in Assumption 2. The rest of the arguments follow via Lipschitzness.

### C.2 THEOREM FOR ROGD-OFU

We state the assumptions first for the later theorems. These assumptions are similar to Assumption 1-3 in Wu et al. (2021).

**Assumption 3** $\forall \mathcal{P} \in \{\mathcal{P}^{\text{train}}, \mathcal{P}_1^{\text{test}}, \cdots, \mathcal{P}_T^{\text{test}}\}$, $\text{diag}(C_{f,\mathcal{P}})$ *is differentiable with respect to $f$.*

**Assumption 4** $\forall t \in [T]$, $\ell(g(\cdot; f_t'', p/q_0); \mathcal{P}_t^{\text{test}})$ *is convex in $p$, where $f_t''$ is defined in Algorithm 1.*

**Assumption 5** $\sup_{p \in \Delta^{K-1}, i \in [K], t \in [T]} \|\nabla_p \ell(g(\cdot; f_t'', p/q_0); \mathcal{P}_t^{\text{test}})\|_2$ *is finite and bounded by $L$.*

**Theorem 3 (Regret convergence for ROGD-OFU)** *If we run Algorithm 1 with ROGD-R (Algorithm 3) and $\eta = \sqrt{\frac{2}{T}}\frac{1}{L}$, under Assumption 3, 4, 5, ROGD-OFU satisfies the guarantee*

$$\mathbb{E}\left[\frac{1}{T}\sum_{t=1}^{T}\ell(f_t^{\text{ogd-ofu}}; \mathcal{P}_t^{\text{test}})\right] - \min_{p \in \Delta_K}\mathbb{E}\left[\frac{1}{T}\sum_{t=1}^{T}\ell(g(\cdot; f_t'', p/q_0); \mathcal{P}_t^{\text{test}})\right] \leq \sqrt{\frac{2}{T}}L. \quad (12)$$

$$\mathbb{E}\left[\frac{1}{T}\sum_{t=1}^{T}\ell(f_t^{\text{ogd}}; \mathcal{Q}_t)\right] - \min_{p \in \Delta_K}\mathbb{E}\left[\frac{1}{T}\sum_{t=1}^{T}\ell(g(\cdot; p, f_0, q_0); \mathcal{Q}_t)\right] \leq \sqrt{\frac{2}{T}}L. \quad (13)$$

**Proof:** For any fixed $p$,

$$\ell(f_t^{\text{rogd-ofu}}; \mathcal{P}_t^{\text{test}}) - \ell(g(\cdot; f_t'', p/q_0); \mathcal{P}_t^{\text{test}}) = \ell(g(\cdot; f_t'', p_t/q_0); \mathcal{P}_t^{\text{test}}) - \ell(g(\cdot; f_t'', p/q_0); \mathcal{P}_t^{\text{test}})$$

$$\leq (p_t - p) \cdot \nabla_p \ell(g(\cdot; f_t'', p_t/q_0); \mathcal{P}_t^{\text{test}})$$

$$= (p_t - p) \cdot J_{p,f_t''}(p_t)^{\top}\mathbb{E}_{S_t}[s_t|S_1, \cdots, S_{t-1}]$$

$$= \mathbb{E}_{S_t}[(p_t - p) \cdot J_{p,f_t''}(p_t)^{\top}s_t|S_1, \cdots, S_{t-1}],$$

where the last inequality holds by the fact that $(p_t - p) \cdot J_{p,f_t''}(p_t)^{\top}$ is independent of $\{S_1, \cdots, S_{t-1}\}$. To bound $(p_t - p) \cdot J_{p,f_t''}(p_t)^{\top}s_t$,

$$\|p_{t+1} - p\|_2^2 = \|\text{Prof}_{\Delta^{K-1}}(p_t - \eta \cdot J_{p,f_t''}(p_t)^{\top}s_t) - p\|_2^2$$

$$\leq \|p_t - \eta \cdot J_{p,f_t''}(p_t)^{\top}s_t - p\|_2^2$$

$$= \|p_t - p\|_2^2 + \eta^2\|J_{p,f_t''}(p_t)^{\top}s_t\|_2^2 - 2\eta(p_t - p) \cdot (J_{p,f_t''}(p_t)^{\top}s_t).$$

This implies

$$(p_t - p) \cdot (J_{p,f_t''}(p_t)^{\top}s_t) \leq \frac{1}{2\eta}(\|p_t - p\|_2^2 - \|p_{t+1} - p\|_2^2) + \frac{\eta}{2}\|J_{p,f_t''}(p_t)^{\top}s_t\|_2^2$$

Thus

$$\mathbb{E}_{S_1, \cdots, S_T}\left[\frac{1}{T}\sum_{t=1}^{T}\ell(f_t^{\text{rogd-ofu}}; \mathcal{P}_t^{\text{test}}) - \frac{1}{T}\sum_{t=1}^{T}\ell(g(\cdot; f_t'', p/q_0); \mathcal{P}_t^{\text{test}})\right]$$

$$\leq \mathbb{E}_{S_1, \cdots, S_T}\left[\frac{1}{T}\sum_{t=1}^{T}\frac{1}{2\eta}(\|p_t - p\|_2^2 - \|p_{t+1} - p\|_2^2) + \frac{\eta}{2}\|J_{p,f_t''}(p_t)^{\top}s_t\|_2^2\right]$$

$$\leq \frac{1}{2\eta T}\|p_1 - p\|_2^2 + \frac{\eta}{2T}\sum_{t=1}^{T}\mathbb{E}_{S_1, \cdots, S_t}[\|J_{p,f_t''}(p_t)^{\top}s_t\|_2^2]$$

$$\leq \frac{1}{\eta T} + \frac{\eta L^2}{2} = \sqrt{\frac{2}{T}}L.$$

This bound holds for any p. Thus,

$$\mathbb{E}_{S_1,\cdots,S_T}\left[\frac{1}{T}\sum_{t=1}^{T}\ell(f_t^{\text{rogd-ofu}};\mathcal{P}_t^{\text{test}})\right] - \min_{p\in\Delta^{K-1}}\mathbb{E}_{S_1,\cdots,S_T}\left[\frac{1}{T}\sum_{t=1}^{T}\ell(g(\cdot;f_t'',p/q_0);\mathcal{P}_t^{\text{test}})\right] \leq \sqrt{\frac{2}{T}}L.$$

### C.3 THEOREM FOR FTH-OFU

We begin with two assumptions.

**Assumption 6** *For any $\mathcal{P}^{\text{test}}$ s.t. $\mathcal{P}^{\text{test}}(x|y) = \mathcal{P}^{\text{train}}(x|y)$, denote $q_t := (\mathcal{P}_t^{\text{test}}(y = k) : k \in [K])$ and then*

$$\|q_t - \arg\min_{p\in\Delta^{K-1}}\ell(g(\cdot;f_t'',p/q_0);\mathcal{P}^{\text{test}})\| \leq \delta.$$

**Assumption 7** $\forall\mathcal{P}^{\text{test}}$ *s.t.* $\mathcal{P}^{\text{test}}(x|y) = \mathcal{P}^{\text{train}}(x|y)$, $\sup_p\|\nabla_p\ell(g(\cdot;f_t'',p/q_0);\mathcal{P}^{\text{test}})\| \leq L$

**Theorem 4 (Regret convergence for FTH-OFU)** *If we run Algorithm 1 with FTH-R (Algorithm 4) and assume $\sigma$ is no larger than the minimum singular value of invertible confusion matrices $\{C_{f_1'',D_0'},\cdots C_{f_T'',D_0'}\}$, under Assumption 6 and 7 with $\delta = 0$, FTH-OFU satisfies the guarantee that with probability at least $1 - 2KT^{-7}$ over samples $S_1\cup\cdots\cup S_T$,*

$$\frac{1}{T}\sum_{t=1}^{T}\ell(f_t^{\text{fth-ofu}};\mathcal{P}_t^{\text{test}}) - \min_{p\in\Delta_K}\frac{1}{T}\sum_{t=1}^{T}\ell(g(\cdot;f_t'',p/q_0);\mathcal{P}_t^{\text{test}}) \leq O\left(\frac{\log T}{T} + \frac{1}{\sigma}\sqrt{\frac{K\log T}{T}}\right),$$
(14)

*where $K$ is the number of classes.*

**Proof:** Denote $q_t := (\mathcal{P}_t^{\text{test}}(y = k) : k \in [K])$. By the Hoeffding and union bound, we have

$$\mathbb{P}\left(\forall t \leq T, \|p_{t+1} - \frac{1}{t}\sum_{\tau=1}^{t}q_\tau\| \leq \sqrt{K}\varepsilon_t\right) \geq 1 - \sum_{t=1}^{T}2M\exp\left(-2\varepsilon_t^2 t/\sigma^2\right).$$

This implies that with probability at least $1 - \sum_{t=1}^{T}2M\exp\left(-2\varepsilon_t^2 t/\sigma^2\right)$, $\forall p$,

$$\sum_{t=1}^{T}\ell(p_t;\mathcal{P}_t^{\text{test}}) - \sum_{t=1}^{T}\ell(g(\cdot;f_t'',p/q_0);\mathcal{P}_t^{\text{test}})$$

$$\leq \sum_{t=1}^{T}\ell(g(\cdot;f_t'',\frac{1}{t}\sum_{\tau=1}^{t}q_\tau/q_0);\mathcal{P}_t^{\text{test}}) - \sum_{t=1}^{T}\ell(g(\cdot;f_t'',p/q_0);\mathcal{P}_t^{\text{test}}) + L\sqrt{M}\cdot\sum_{t=1}^{T}\varepsilon_t$$

$$\leq \sum_{t=1}^{T}\ell(g(\cdot;f_t'',\frac{1}{t-1}\sum_{\tau=1}^{t-1}q_\tau/q_0);\mathcal{P}_t^{\text{test}}) - \sum_{t=1}^{T}\ell(g(\cdot;f_t'',\frac{1}{t}\sum_{\tau=1}^{t}q_\tau/q_0);\mathcal{P}_t^{\text{test}}) + L\sqrt{M}\cdot\sum_{t=1}^{T}\varepsilon_t$$

$$\leq \sum_{t=1}^{T}L\left\|\frac{1}{t-1}\sum_{\tau=1}^{t-1}q_\tau - \frac{1}{t}\sum_{\tau=1}^{t}q_\tau\right\| + L\sqrt{M}\cdot\sum_{t=1}^{T}\varepsilon_t$$

$$\leq \sum_{t=1}^{T}\frac{L}{t}\left\|\frac{1}{t-1}\sum_{\tau=1}^{t-1}q_\tau - q_t\right\| + L\sqrt{M}\cdot\sum_{t=1}^{T}\varepsilon_t$$

$$\leq \sum_{t=1}^{T}\frac{2L}{t} + L\sqrt{M}\cdot\sum_{t=1}^{T}\varepsilon_t.$$

If we take $\varepsilon_t = 2\sigma\sqrt{\frac{\ln T}{T}}$, the above is equivalent to: with probability at least $1 - 2KT^{-7}$

$$\frac{1}{T}\sum_{t=1}^{T}\ell(p_t;\mathcal{P}_t^{\text{test}}) - \min_p\frac{1}{T}\sum_{t=1}^{T}\ell(g(\cdot;f_t'',p/q_0);\mathcal{P}_t^{\text{test}}) \leq 2L\frac{\ln T}{T} + 4L\sigma\sqrt{\frac{K\ln T}{T}}$$

## C.4 THEOREMS FOR UOGD-OFU AND ATLAS-OFU

**Theorem 5** *[Regret convergence for UOGD-OFU] Let $f(\cdot; \theta_{f''_t}^{\text{feat}}, w)$ denote a network with the same feature extractor as that of $f''_t$ and a last linear layer with weight $w$. Let $f^{\text{uogd-ofu}} = f(\cdot; \theta_{f''_t}^{\text{feat}}, w_t)$, where $w_t$ is the weight maintained at round $t$ by Algorithm 5. If we run Algorithm 1 with UOGD in Bai et al. (2022) and let step size be $\eta$, then under the same assumptions as Lemma 1 in Bai et al. (2022), UOGD-OFU satisfies that*

$$\mathbb{E}\left[\frac{1}{T}\sum_{t=1}^{T}\ell(f^{\text{uogd-ofu}}; \mathcal{P}_t^{\text{test}}) - \frac{1}{T}\sum_{t=1}^{T}\min_{w\in\mathcal{W}}\ell(f(\cdot; \theta_{f''_t}^{\text{feat}}, w); \mathcal{P}_t^{\text{test}})\right] \leq O\left(\frac{K\eta}{\sigma^2} + \frac{1}{\eta T} + \sqrt{\frac{V_{T,\ell}}{T\eta}}\right),$$
(15)

*where $V_{T,\ell} := \sum_{t=2}^{T}\sup_{w\in\mathcal{W}}|\ell(f(\cdot; \theta_{f''_t}^{\text{feat}}, w); \mathcal{P}_t^{\text{test}}) - \ell(f(\cdot; \theta_{f''_{t-1}}^{\text{feat}}, w); \mathcal{P}_{t-1}^{\text{test}})|$, $\sigma$ denotes the minimum singular value of the invertible confusion matrices $\{C_{f''_1, D'_0}, \cdots C_{f''_T, D'_0}\}$ and $K$ is the number of classes and the expectation is taken with respect to randomness in the revealed co-variates.*

**Proof Sketch:** Recall that $\ell(f(\cdot; \theta_{f''_t}^{\text{feat}}, w); \mathcal{P}_t^{\text{test}}) := E_{(x,y)\sim\mathcal{P}_t^{\text{test}}}\ell_{\text{ce}}\left(f(x|\theta_{f''_t}^{\text{feat}}, w), y\right)$.

This guarantee follows from the arguments in Bai et al. (2022) from two basic facts below:

1. The risk $\ell(f(\cdot; \theta_{f''_t}^{\text{feat}}, w); \mathcal{P}_t^{\text{test}})$ is convex in $w$ over a convex and compact domain $\mathcal{W}$.

2. It is possible to form unbiased estimates $\hat{G}_t(w) \in \mathbb{R}^K$ such that $E[\hat{G}_t(w)|S_{1:t-1}] = E_{((x,y)\sim\mathcal{P}_t^{\text{test}})}\nabla_w\ell_{\text{ce}}\left(f(x|\theta_{f''_t}^{\text{feat}}, w), y\right)$.

Hence we proceed to verify these two facts in our setup. Fact 1 is true because the cross-entropy loss is convex in any subset of the simplex and the last linear layer weights only defines an affine transformation which preserves convexity.

For fact 2, note that the $f''_t$ only uses the data until round $t - 1$. So by the same arguments in Bai et al. (2022), using the BBSE estimator defined from the classifier $f''_t$, the unbiased estimate of risk gradient can be defined.

Let $w_t$ be the weight of the last layer maintained by UOGD at round t. Let $u_{1:T}$ be any sequence in $\mathcal{W}$. Consequently we have for any round,

$$\ell(f^{\text{uogd-ofu}}; \mathcal{P}_t^{\text{test}}) - \ell(f(\cdot; \theta_{f''_t}^{\text{feat}}, u_t)) = \ell(f(\cdot; \theta_{f''_t}^{\text{feat}}, w_t) - \ell(f(\cdot; \theta_{f''_t}^{\text{feat}}, u_t)) \tag{16}$$

$$\leq \langle\nabla_w\ell(f(\cdot; \theta_{f''_t}^{\text{feat}}, w_t), w_t - u_t\rangle \tag{17}$$

$$= \langle E[\hat{G}_t(w_t)|S_{1:t-1}], w_t - u_t\rangle. \tag{18}$$

Rest of the proof is identical to Bai et al. (2022).

**Theorem 6 (Regret convergence for ATLAS-OFU)** *Let $f(\cdot; \theta_{f''_t}^{\text{feat}}, w)$ denote a network with the same feature extractor as that of $f''_t$ and a last linear layer with weight $w$. Let $f^{\text{atlas-ofu}} = f(\cdot; \theta_{f''_t}^{\text{feat}}, w_t)$, where $w_t$ is the weight maintained at round $t$ by Algorithm 6. If we run Algorithm 1 with ATLAS in Bai et al. (2022) and set up the step size pool $\mathcal{H} = \{\eta_i = O\left(\frac{\sigma}{\sqrt{KT}}\right) \cdot 2^{i-1}|i \in [N]\}$ ($N = 1 + \lceil\frac{1}{2}\log_2(1 + 2T)\rceil$), then under the same assumptions as Lemma 1 in Bai et al. (2022), UOGD-OFU satisfies that*

$$\mathbb{E}\left[\frac{1}{T}\sum_{t=1}^{T}\ell(f^{\text{atlas-ofu}}; \mathcal{P}_t^{\text{test}}) - \frac{1}{T}\sum_{t=1}^{T}\min_{w\in\mathcal{W}}\ell(f(\cdot; \theta_{f''_{t+1}}^{\text{feat}}, w); \mathcal{P}_t^{\text{test}})\right] \leq O\left(\left(\frac{K^{1/3}}{\sigma^{2/3}} + 1\right)\frac{V_{T,\ell}^{1/3}}{T^{1/3}} + \sqrt{\frac{K}{\sigma^2 T}}\right),$$
(19)

*where $V_{T,\ell} := \sum_{t=2}^{T}\sup_{w\in\mathcal{W}}|\ell(f(\cdot; \theta_{f''_t}^{\text{feat}}, w); \mathcal{P}_t^{\text{test}}) - \ell(f(\cdot; \theta_{f''_{t-1}}^{\text{feat}}, w); \mathcal{P}_{t-1}^{\text{test}})|$, $\sigma$ denotes the minimum singular value of the invertible confusion matrices $\{C_{f''_1, D'_0}, \cdots C_{f''_T, D'_0}\}$ and $K$ is the number of classes and the expectation is taken with respect to randomness in the revealed co-variates.*

The proof is similar to that of Theorem 5 and hence omitted.

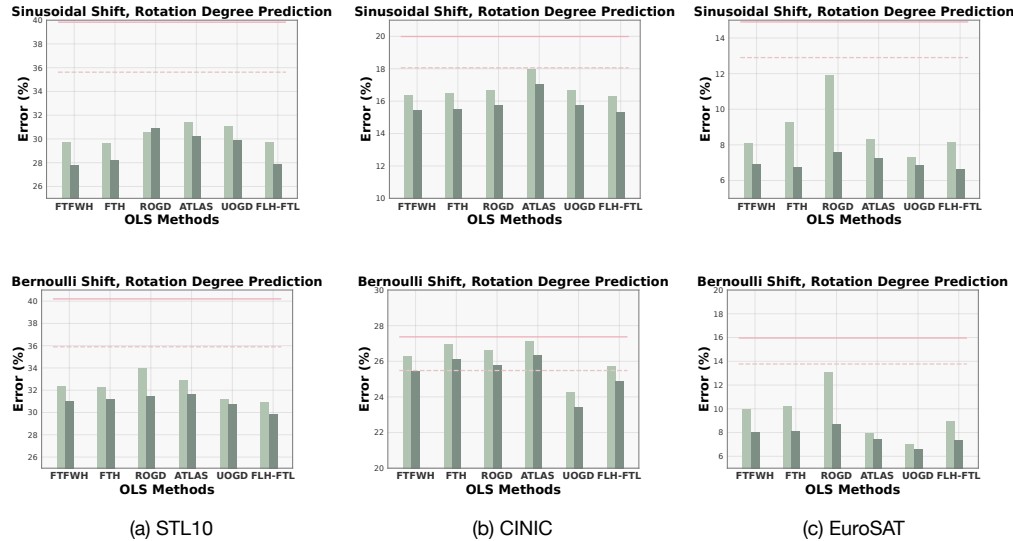

Figure 5: Results for additional datasets STL10, CINIC and EuroSAT.

**Discussion about the assumption.** In the theorems for UOGD and ATLAS, the definition of $V_{T,\ell}$ is shift severity from $\mathcal{P}_t^{\text{test}}$. However, in the theorems for UOGD-OFU and ATLAS-OFU above, $V_{T,\ell}$ is shift severity from both $\mathcal{P}_t^{\text{test}}$ and $\theta_{f_t''}^{\text{feat}}$, which can be much larger. This might lead to harder convergence of the regret.

## D  ADDITIONAL EXPERIMENTS

### D.1  ADDITIONAL DETAILS OF DATASETS

**Severity of CIFAR-10C in the experiment.** For each type of corruption in CIFAR-10C, we select a mild level and a high level of severity in the experiment section. Here we introduce the exact parameters of mild and high levels of severity for those corruptions. For Gaussian Noise, the severity levels for [mild, high] are [0.03, 0.07]. For Fog, the severity levels for [mild, high] are [(0.75,2.5), (1.5,1.75)]. For Pixelate, the severity levels for [mild, high] are [0.75, 0.65].

**Details of additional datasets.** In addition to CIFAR-10 and CIFAR-10C, we evaluate on three more datasets: STL10 (Coates et al., 2011), CINIC (Darlow et al., 2018), and EuroSAT (Helber et al., 2019). Similar to CIFAR-10, we split the original train sets of these datasets into the train set and the validation set by the ratio 4 : 1 and use the original test sets for sampling test images in the online test stage.

### D.2  RESULTS ON ADDITIONAL DATASETS

Figure 5 shows the results for three additional datasets: STL10, CINIC, and EuroSAT, also with OLS-OFU (rotation degree prediction) under sinusoidal shift setting. In-line with CIFAR-10, we observe that OLS and OLS-OFU can perform better than Base and OFU, and OLS-OFU can outperform OLS. We find this pattern to be more consistent for EuroSAT and STL10 than CINIC.

### D.3  MORE RESULTS ON CIFAR-10

Figure 6 shows the results on CIFAR-10 for Bernoulli shift cross three SSL methods in OLS-OFU. Similar to Figure 2(a), OLS-OFU within all three SSL methods outperforms all baseline methods.

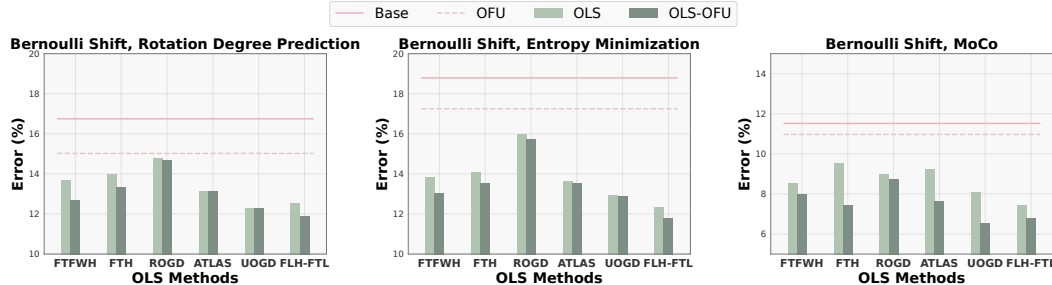

Figure 6: Results of Bernoulli shifts on CIFAR-10. OLS-OFU is evaluated with three SSL methods.

## D.4 MORE RESULTS ON CIFAR-10C

We evaluate three SSL methods in OLS-OFU on CIFAR-10C for two online shift patterns. We pick moderately high severity levels for evaluating CIFAR-10C. In Figure 7 we can observe the consistent improvement from OLS to OLS-OFU and the SOTA performance of OLS-OFU.

## D.5 MORE RESULTS ON CIFAR-10C WITH HIGH SEVERITY

We evaluate three SSL methods in OLS-OFU on CIFAR-10C with *high severity* for two online shift patterns. In Figure 8, we can observe when SSL in OLS-OFU is rotation degree prediction or MoCo, the improvement from OLS to OLS-OFU is very significant but OLS-OFU cannot outperform OFU. The conclusion is similar to the discussion for Figure 3 in Section 4. However, OLS-OFU with SSL entropy minimization has different behavior. When the corruption of CIFAR-10C becomes more severe, OLS-OFU entropy minimization shows less improvement from OLS, if we compare the transition: Figure 7 (clean CIFAR-10), Figure 7 (CIFAR-10C with mild severity), Figure 8 (CIFAR-10C with high severity). This suggests that rotation degree prediction and MoCo are more appropriate SSL to address the domain shift from CIFAR-10 to CIFAR-10C.

## D.6 EMPIRICAL EVALUATION FOR EQUATION 5

Figure 9 shows the comparison between LHS (OLS-OFU) and RHS (OLS) of the inequality in Equation 5 when the SSL in OLS-OFU is entropy minimization or MoCo. Similar to what we observe in Figure 4, the inequality in Equation 5 holds cross 4 data settings and two online shift patterns when OLS-OFU is implemented with entropy minimization or MoCo.

## D.7 ABLATION FOR THE ORDER OF UPDATES AND PREDICTIONS

In the default framework of online distribution shift as shown in Figure 1, the model updates happen after the predictions for the samples at time step $t$. We would like to see if the model updated before the predictions would bring benefit. Figure 10 shows the comparison between "predict first" and "update first". We can observe that there is no strong evidence to demonstrate the advantage of "predict first" or "update first" – the difference is indeed insignificant. However, because in the framework of "predict first" OLS and OLS-OFU enjoy strong theoretical guarantees, we recommend "predict first" in practice.

## D.8 DETAILS OF SSL METHODS

When the SSL loss is rotation degree prediction, it requires another network $f^{\deg}$ to predict the rotation degree, sharing the same feature extractor $\theta^{\text{feat}}$ as $f_0$ but with a different set of downstream layers. Its SSL loss $\ell_{\text{ssl}}(S; f)$ is defined as $\sum_{x \in S} \ell_{ce}(f^{\deg}(R(x, i)), i)$, where $i$ is an integer uniformly sampled from $[4]$, and $R(x, i)$ is to rotate $x$ with degree $\text{DL}[i]$ from a list of degrees $\text{DL} = [0, 90, 180, 270]$. Alternatively, if the SSL loss is entropy minimization, $\ell_{\text{ssl}}(S; f)$ would be the entropy $\sum_{x \in S} \sum_{k=1}^{K} f(x)_k \log f(x)_k$. Moreover, the SSL loss would be a contrastive loss (In-foNCE) where the positive example $x'$ is an augmented version of $x$ and other samples in the same

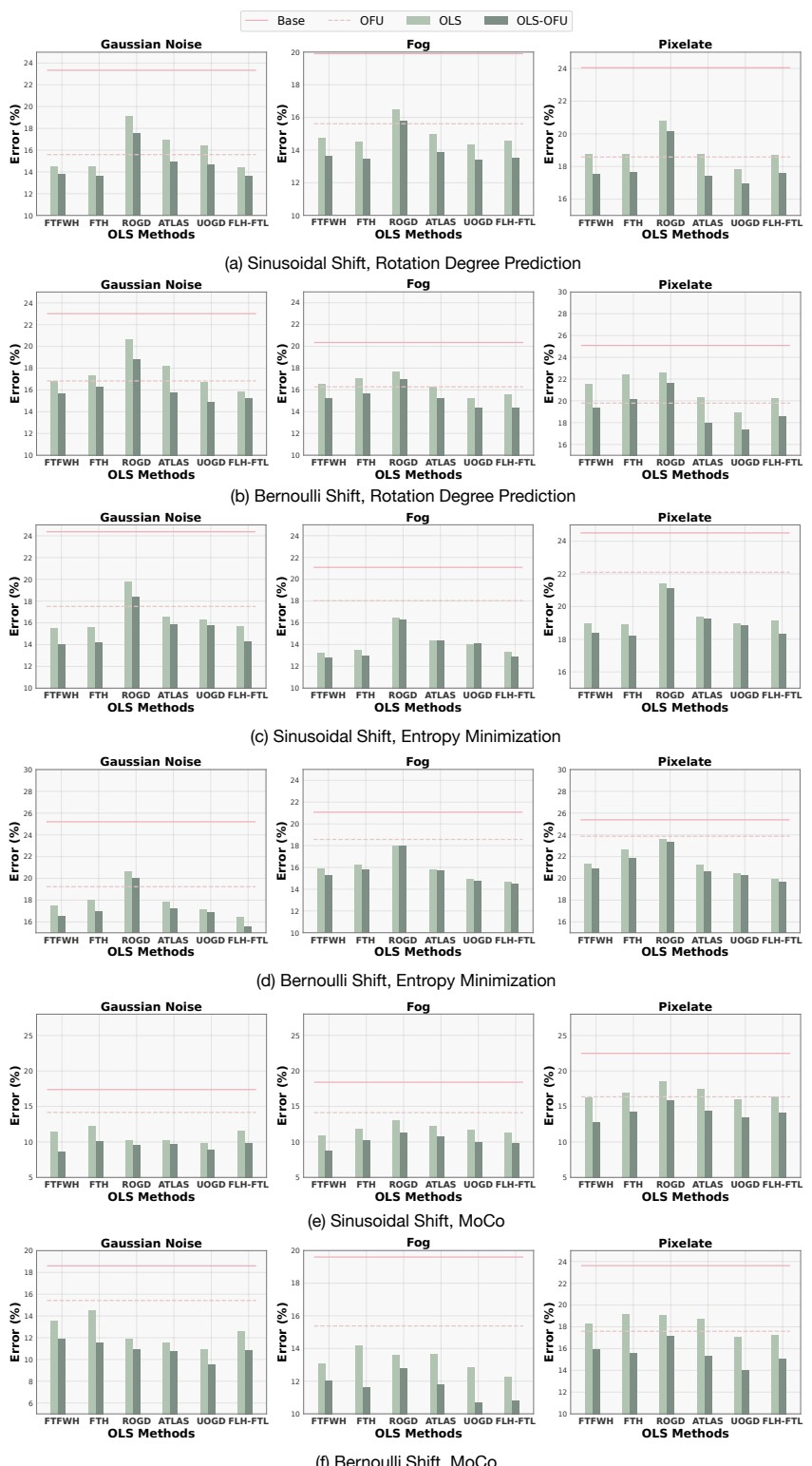

Figure 7: Results of two online shift patterns on CIFAR-10C and three SSL methods in OLS-OFU.

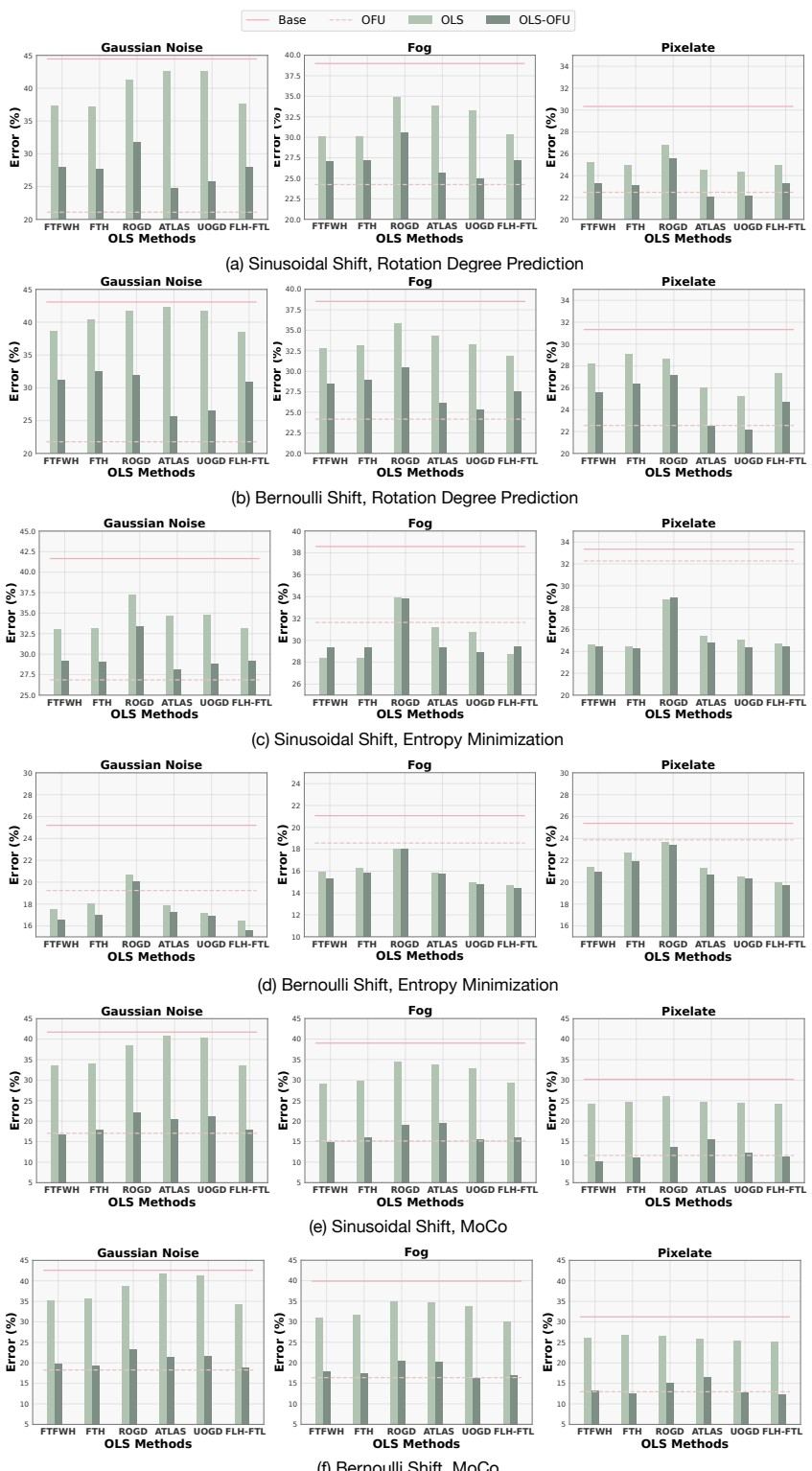

Figure 8: Results of two online shift patterns on CIFAR-10C (high severity) and three SSL methods in OLS-OFU.

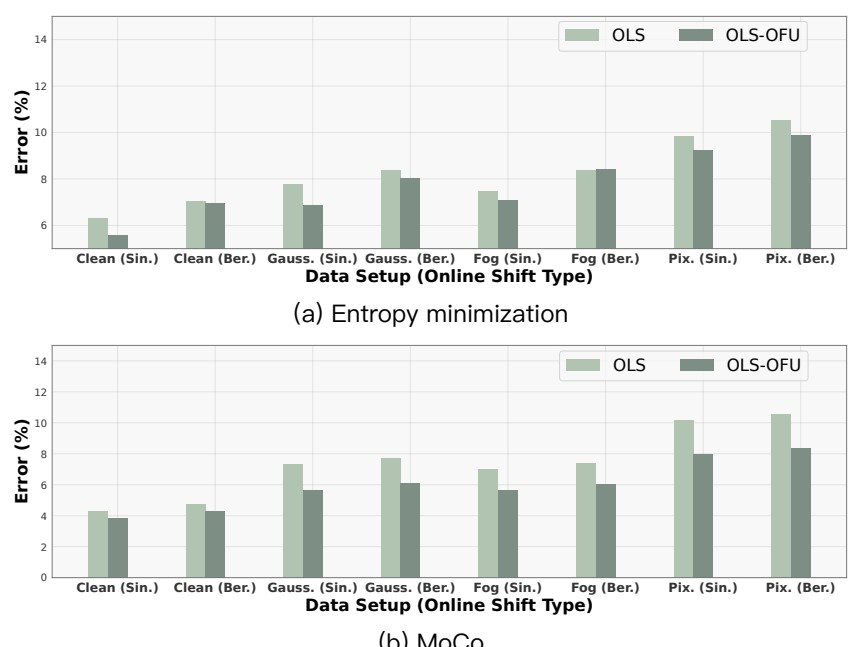

(a) Entropy minimization

(b) MoCo

Figure 9: Sanity check for the inequality $\mathbb{E}\left[\frac{1}{T}\sum_{t=1}^{T}\ell(g(\cdot;f_t'',q_t/q_0);\mathcal{P}_t^{\text{test}})\right] < \frac{1}{T}\sum_{t=1}^{T}\ell(g(\cdot;f_0,q_t/q_0);\mathcal{P}_t^{\text{test}})$ (Equation 5) when SSL in OLS-OFU is entropy minimization or MoCo.

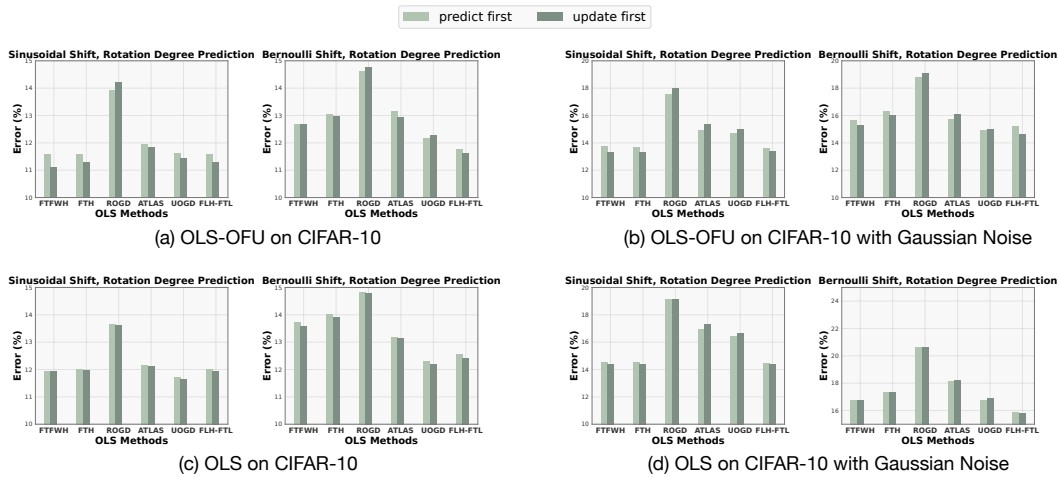

Figure 10: Ablation for the order of updates and predictions.

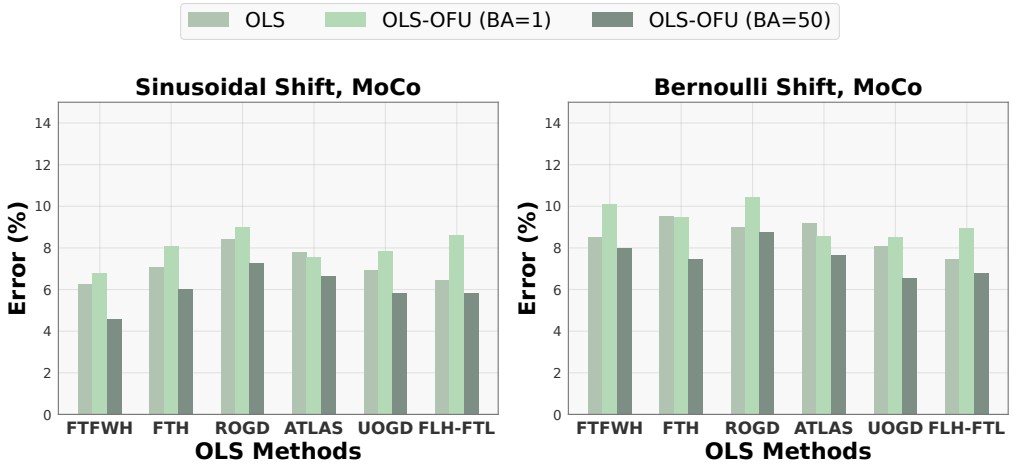

Figure 11: Evaluating OLS-OFU with different batch accummulations for MoCo on CIFAR-10.

time step can be the negative examples. However, the batch size for a time step is small, e.g. 10 in our experiment. MoCo updates with such a small batch won't work. Thus, we experiment with MoCo by applying a batch accumulation strategy, which we introduce next.

### D.8.1  BATCH ACCUMULATION FOR MOCO

Prior test-time training (rotation degree prediction, entropy minimization) methods operate with the OFU framework easily, with feature extractor updates taking place in each time step. Rotation degree prediction originates as a self-supervised training method (Gidaris et al., 2018), thus naturally we evaluate more recent self-supervised training methods, namely MoCo (He et al., 2019; Chen et al., 2020b; 2021). No prior work shows how to use MoCo (or self-supervised learning in general) in a test-time training setting. Given that self-supervised training is sensitive to batch size, the intuition is that a larger batch size (much larger than the number of online samples available per time step) is required to perform a valid gradient update for a MoCo checkpoint. As such, we evaluated a *batch accumulation* strategy OLS-OFU (BA=$\tau$), where we continue evaluating online samples per time step, but only perform the online feature update at every $\tau$ steps (having accumulated the online samples throughout the $\tau$ steps in one batch). In particular, we perform feature extractor update every $\tau = 50$ steps (for 1000 steps, feature update occurs 20 times, online samples evaluation occurs 1000 times), evaluating with 10 online samples per time step, using a smaller learning rate (0.0005) but test-time train with 10 epochs. Notice that $\tau = 1$ is the default setting in Algorithm 1. OLS-OFU with MoCo presented in Figure 2, Figure 6 and Figure 7 is equivalent to OLS-OFU (BA=50).

To show the necessity of large $\tau$, we evaluate both OLS-OFU (BA=1) and OLS-OFU (BA=50) on CIFAR-10 (Figure 11) and CIFAR-10C (Figure 12). Firstly, we can observe that OLS-OFU (BA=1) is even worse than OLS. We hypothesize this is because small batch size of MoCo will hurt the performance and larger batch size in MoCo is necessary. Hence, we increase $\tau$ from 1 to 50 and then we can observe the significant improvement from OLS-OFU (BA=1) to OLS-OFU (BA=50). Now, OLS-OFU (BA=50) can outperform OLS.

### D.9  IMPROVEMENTS COMPARED WITH LITERATURE

We would like to compare the improvements in the literature for the OLS problem and the improvements from the SOTA in the literature to our method OLS-OFU. We can calculate the difference $\Delta_{\text{error}}$ between the errors of method A and B to show the improvement $A \to B$. To measure the improvements in the literature for the OLS problem, we measure $\Delta_{\text{error}}$ between FTH (proposed

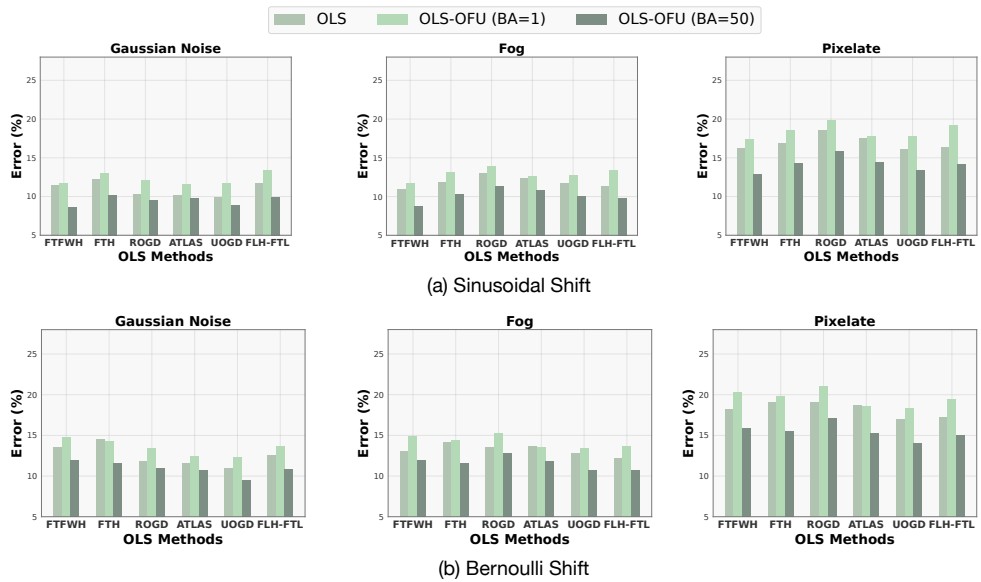

Figure 12: Evaluating OLS-OFU with different batch accummulations for MoCo on CIFAR-10C.

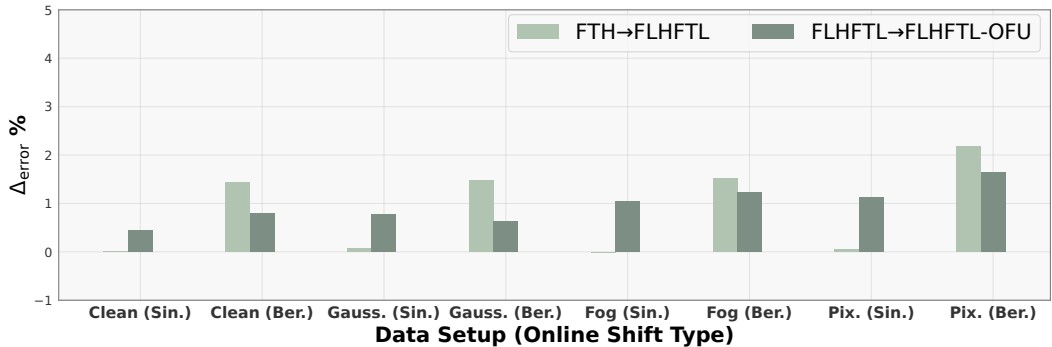

Figure 13: Improvements from FTH to FLHFTL and improvements from FLHFTL to FLHFTL-OFU across all datasets and online shift patterns.

in the first OLS paper) and FLHFTL (SOTA in the literature). We also calculate $\Delta_{\text{error}}$ between FLHFTL and FLHFTL-OFU.

Figure 13 shows the $\Delta_{\text{error}}$ for FTH→FLHFTL and FLHFTL→FLHFTL-OFU across all data settings (dataset and online shift pattern) when the SSL in $OFU$ is set as rotation degree prediction. The improvement of FLHFTL→FLHFTL-OFU is as significant as the improvement of FTH→FLHFTL. Moreover, the improvement of FLHFTL→FLHFTL-OFU is more consistent. This consistency demonstrates the potential that OLS-OFU can have improvement for any future OLS algorithm.

## D.10 SELF-TRAINING

Pseudo labelling (Lee, 2013), a common self-training technique, generates pseudo labels for unlabelled data and uses them to update the model. Though we are not able to use ground-truth labels to compute feature extractor updates, we can use the model at time $t$ to make predictions with respect to the online samples at time $t$, and train on the inputs with their assigned (pseudo) labels. An issue that arises in self-training is confirmation bias, where the model repeatedly overfits to incorrect pseudo-labels. As such, different methods can be used to select which samples will be pseudo-labelled and used in updating the model, e.g. using data augmentation (Arazo et al., 2020), using

regularization to induce confident low-entropy pseudo-labelling (Grandvalet & Bengio, 2004a), using softmax thresholds to filter out noisy low-confidence predictions (Xie et al., 2020). We make use of ensembles to identify noisy low-confidence/entropy pseudo-label predictions, though other various alternatives can also be used. In addition to OLS and OLS-OFU, we highlight the methods under comparison:

- *OLS-OFU ($\ell_{\text{sup}}(\cdot, y_{\text{ground-truth}})$)*: Instead of computing pseudo-labels, we make use of the correct ground-truth labels $y_{\text{ground-truth}}$. Recall $\ell_{\text{sup}}$ is the supervised learning loss. We update the feature extractor with the supervised loss w.r.t. ground-truth labels $\ell_{\text{sup}}(\cdot, y_{\text{ground-truth}})$.

- *OLS-OFU ($\ell_{\text{ssl}} + \ell_{\text{sup}}(\cdot, y_{\text{ground-truth}})$)*: Instead of computing pseudo-labels, we make use of the correct ground-truth labels $y_{\text{ground-truth}}$. Recall $\ell_{\text{ssl}}$ and $\ell_{\text{sup}}$ are the self-supervised and supervised learning losses respectively. We update the feature extractor with both the self-supervised loss $\ell_{\text{ssl}}$ as well as the supervised loss w.r.t. ground-truth labels $\ell_{\text{sup}}(\cdot, y_{\text{ground-truth}})$.

- *OLS-OFU ($\ell_{\text{ssl}} + \ell_{\text{sup}}(\cdot, y_{\text{pseudo-label(\#samples=, \#FU-samples=)}})$)*: Recall $\ell_{\text{ssl}}$ and $\ell_{\text{sup}}$ are the self-supervised and supervised learning losses respectively. We compute pseudo-labels $y_{\text{pseudo-label}}$), and update the feature extractor with both the self-supervised loss $\ell_{\text{ssl}}$ as well as the supervised loss w.r.t. pseudo-labels $\ell_{\text{sup}}(\cdot, y_{\text{pseudo-label}})$.

**How to compute pseudo-labels?** We now describe the procedure to compute pseudo-labels for $\ell_{\text{sup}}(\cdot, y_{\text{pseudo-label(\#samples=, \#FU-samples=)}})$. The seed used to train our model is 4242, and we train an additional 4 models on seeds 4343, 4545, 4646, 4747. With this ensemble of 5 models, we keep sampling inputs at each online time step until we have `#FU-samples` samples, or we reach a limit of `#samples` samples. We accept an input when the agreement between the ensembles exceeds a threshold $e = 1.0$ (i.e. we only accept samples where all 5 ensembles agree on the label of the online sample). In the default online learning setting, there are only `#samples=10`, and therefore there may not be enough accepted samples to perform feature update with, thus we evaluate with a continuous sampling setup, where we sample `#samples=50` (and evaluate on all these samples), but only use the first 10 samples (`#FU-samples=10`) to perform the feature extractor update.

**Results on pseudo-labelling.** First, we find that *OLS-OFU ($\ell_{\text{ssl}} + \ell_{\text{sup}}(\cdot, y_{\text{ground-truth}})$)* attains the lowest error and is the lower bound we are attaining towards. Evaluating *OLS-OFU ($\ell_{\text{ssl}} + \ell_{\text{sup}}(\cdot, y_{\text{pseudo-label(\#samples=10, \#FU-samples=10)}})$)*, we find that the performance does not outperform OLS-OFU, and is not near *OLS-OFU ($\ell_{\text{ssl}} + \ell_{\text{sup}}(\cdot, y_{\text{ground-truth}})$)*. If we set the threshold $e$ too high, there may not be enough online samples to update the feature extractor. If we set the threshold $e$ too low, there may be too many incorrect labels and we incorrectly update our feature extractor. As such, we would like to sample more inputs at each online time step such that we can balance this tradeoff. We sample `#samples=50` at each online time step, and update with `#FU-samples` $\leq$ `10`. For fair comparison, we also show the comparable methods in both `#samples=10, #FU-samples=10` and `#samples=50, #FU-samples=50` settings.

With this sampling setup, we find that *OLS-OFU ($\ell_{\text{ssl}} + \ell_{\text{sup}}(\cdot, y_{\text{pseudo-label(\#samples=50, \#FU-samples=10)}})$)* can outperform both *OLS-OFU (`#samples=10`)* and *OLS-OFU (`#samples=50`)*. Though it does not exceed neither *OLS-OFU ($\ell_{\text{ssl}} + \ell_{\text{sup}}(\cdot, y_{\text{ground-truth}})$)* for `#samples=10` nor `#samples=50`, it lowers the gap considerably.

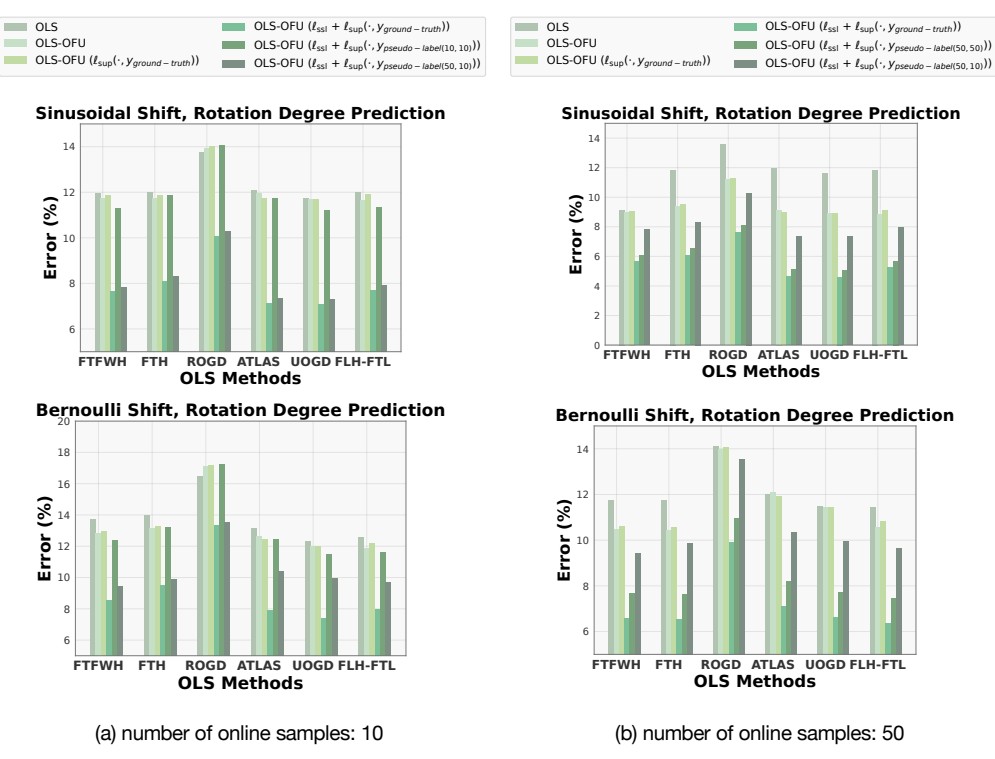

Figure 14: Results on pseudo-labelling.

