# OpenReview forum: "Online Feature Updates Improve Online (Generalized) Label Shift Adaptation"
_ICLR.cc/2024/Conference — Submitted to ICLR 2024_

### Official Review · Reviewer_5U9r · 2023-10-29

**Soundness:** 2 fair
**Presentation:** 2 fair
**Contribution:** 1 poor
**Rating:** 3
**Confidence:** 3

**Summary:**

This paper considers the label shift between training data and testing data, and focuses on the online case, where several batches of testing data arrive sequentially. To this end, the authors propose a method called Online Label Shift adaptation with Online Feature Updates (OLS-OFU), the main idea of which is to utilize a self-supervised learning (SSL) technique to refine the feature extraction process of existing OLS algorithms. Experimental results are presented to verify the performance of their method.

**Strengths:**

1) Label shift is a common phenomenon in real applications, especially those with streaming data. So, the motivation of this study is clear.
2) It seems that previous studies mainly focused on the online label shift (OLS) problem, and this paper is the first one to study the online generalized label shift problem, which introduces an unknown mapping $h(\cdot)$ between the original data to some latent spaces.
3) It seems that this paper is the first one to utilize the self-supervised learning (SSL) technique to improve the feature extraction process of existing OLS algorithms, which can further take advantage of unlabeled data to improve the testing performance.

**Weaknesses:**

1) The proposed method, namely Online Label Shift adaptation with Online Feature Updates (OLS-OFU), is a straightforward combination of any existing self-supervised learning (SSL) technique and any existing OLS algorithm, the novelty of which is limited. Moreover, it seems that there does not exist any challenge in this combination.
2) Although the authors provide some theoretical guarantees for the proposed method, it seems that these results can be simply derived by following previous studies.
3) Although the application of SSL is reasonable, the authors do not provide theoretical guarantees on its performance of learning the implicit feature mapping.

**Questions:**

As discussed in the above weaknesses, the authors should explain the novelty of their method and theoretical results. Moreover, the authors should also discuss the theoretical guarantees of SSL.

---

> ### Author Response · Authors · 2023-11-16
> **Official Comment by Submission7196 Authors**
>
> Thank you for your constructive feedback!
>
> **Novelty and contributions**: Please see our general reply “Clarifications on Contribution”.
>
> **Theoretical guarantee of implicit feature mapping**: Theoretical analysis on domain adaptation/feature learning is generally hard in the context of deep learning. Instead, we varied the severity of data shifts in the experiment and studied how OLS-OFU is influenced. Figure 2(b) shows the results under a mild severity of data shifts. Because the feature extractor is refined for the shift distribution and the violation of the label shift assumption is alleviated, OLS-OFU still benefits from the OLS module if we compare OLS-OFU with OFU. However, when the data is shifted with a higher level of severity, after the same feature refining, the violation of the label shift assumption cannot be controlled under a certain level so the OLS-OFU should not benefit from the OLS module. This is validated by the empirical results: as shown in Figure 3, OLS-OFU performs worse than the single OFU without any OLS module.

---

### Official Review · Reviewer_xmwo · 2023-10-29

**Soundness:** 2 fair
**Presentation:** 3 good
**Contribution:** 2 fair
**Rating:** 5
**Confidence:** 3

**Summary:**

The paper study the problem of online label shift adaptation. Different from existing methods that focus on adjusting or updating the final layer of the pre-trained model, the paper proposes to update the feature representation layers by using self-supervised learning techniques.

**Strengths:**

1. The motivation of this paper is clear. Considering that most of existing methods only consider updating the final layer of classifier, the paper uses the self-supervised learning techniques to learn good feature representations and improve model performance.

2. The implementation is very simple and easy to follow.

**Weaknesses:**

1. The idea is not very novel. Considering that self-supervised learning techniques are originally designed to boost feature representation learning, the idea presented in this paper does not appear very novel. Improving model performance by incorporating self-supervised techniques seems intuitive and not a very surprising and insightful finding.

2. The technical contribution is limited. This paper hardly introduces any new technique the proposed method is merely a combination of existing techniques.

3. The theoretical results in the paper seem to be derived from existing works.

4. Experiments are weak in the current version.

1) Only the results from one dataset are reported in the main paper. These results are insufficient to validate the effectiveness of the proposed method.

2) The experimental results presented in the main paper provide limited information and lack many ablation experiments, which are crucial to supporting the conclusions of the paper. For example, how self-supervised learning techniques improve the model performance?

**Questions:**

How does self-supervised learning work in the studied learning scenarios? Is there any difference from the original way it works?

---

> ### Author Response · Authors · 2023-11-16
> **Official Comment by Submission7196 Authors**
>
> Thank you for your constructive feedback!
>
> **Novelty and technical contribution**: Please see our general reply “Clarifications on Contribution” about the novelty and technical contribution.
>
> **Results presentation in the main paper**: Thank you for the suggestions. The results on several other datasets (STL10, CINIC, EuroSAT) are presented in Figure 5 and they actually show the same improvement pattern as the results for CIFAR10 (Figure 2). We agree that moving these results to the main paper will make the conclusion look more convincing. We will fix the pages in the revision.
>
> **Ablation study**: Thank you for raising this concern. We actually have done the main ablations for our method implied by the main results (Figure 2). We explain these ablations below:
> - The ablation for the OFU module. The OFU module in OLS-OFU can be studied by comparing OLS-OFU and OLS in the main result Figure 2. We can observe consistent improvement across all OLS methods, which demonstrates the advantage of the OFU module introduced in this paper.
> - The ablation for the OLS module. The OLS module in OLS-OFU can be studied by comparing OLS-OFU and OFU in the main result Figure 2. The improvement of OLS-OFU over OFU shows the superiority of the OLS module in the OLS problem.
>
> We will highlight these comparisons in the revision. Moreover, in the last paragraph of the experimental section, we also studied whether to "make predictions" first or "update $f_t$ to $f_{t+1}$" first as an ablation study for the overall framework of OLS algorithms (as shown in Figure 1).

---

### Official Review · Reviewer_VDvw · 2023-10-30

**Soundness:** 2 fair
**Presentation:** 3 good
**Contribution:** 2 fair
**Rating:** 5
**Confidence:** 3

**Summary:**

This paper investigates the problem of label shift in the online setting, where the data distributions vary in a non-stationary environment. Unlike the previous which focused on re-weighting the pretrained classifier or re-training the final linear layer of the classifier, this paper proposed to enhance the feature representation by leveraging the unlabeled data at test time. Specifically, a novel Online Label Shift (OLS) adaptation with Online Feature Updates (OFU), named OLS-OFU, was proposed by refining the feature extraction process through self-supervised learning (SSL). Theoretical analysis indicated that the proposed OLS-OFU could reduce the algoritmic regret by introducing the SSL. And the empirical results showed the effectiveness and the robustness of the proposed method under both the OLS setting and the online generalized label shift (OGLS) setting.

**Strengths:**

1. The setting that this paper considered, online (generalized) label shift, is interesting and realistic in the practical scenarios. The investigation of the adaptation methods in such scenarios is valuable.
2. The method proposed in this work is simple. It can be flexibly combined with different existing online label shift adaptation methods, such as FLHFTL, ROGD, UOGD, etc.
3. Most parts of this paper are well-written and easy to follow.

**Weaknesses:**

1. In my opinion, the motivation of introducing feature extraction refinement is not strong. In the Introduction part, the authors claimed that "feature extractors can still be improved, even during test-time and in the absence of labels". And the authors hypothesized that "a similar effect can be leveraged in the generalized label shift setting". However, I did not see strong relations between the drawbacks of the existing methods in this field and the lack of feature representation improvement.
2. The theoretical support in this work is not well verified by the empirical results. In Eq.(5) of the draft, the math showed that the online updates yield improvements. However, this point cannot be theoretically guaranteed. Furthermore, existing empirical results cannot reveal the relationship between the larger improvement and the better performance.
3. Some algorithmic steps are a little confusing for the readers.


See Questions part for more details.

**Questions:**

1. About the motivation of introducing the feature refinement via self-supervised learning. I did not clearly get why we should focus on the feature improvement. For example, why did you believe the feature representation learning has not been enough in the existing methods? Why should we introduce the feature representation improvement, and is there any empirical support? If we cannot explain this point, introducing self-supervised learning just looks like a naive combination of the online label shift adaptation with the popular topic, self-supervised learning.
2. About the theoretical analysis in Eq. (5). I believe that the motivation of introducing the online feature updates comes from the assumption in this inequality. 5And the authors tried to provide empirical evaluations in Sec. 4 (and Appendix D.6) to verify the holdness of this inequality. However, there is no further demonstrations of the quantitative relationship between the amount of feature representation refinement (or in other words the tightness of Eq. (5)) and the final performance. I will describe my question in the mathematical way:

Suppose $\mathcal{M}$ denote the OLS methods (e.g., $\mathcal{M}\in${FLHFTL, FTH, ROGD,...}). Take $\mathcal{M}=$FLHFTL as an example.
Let
$$X^{\mathcal{M}}=\mathbb{E}[ \frac{1}{T} \Sigma_{t=1}^{T}\ell(f_{t}^{\mathcal{M}-ofu}; \mathcal{P}_{t}^{test})]$$

$$Y=\mathbb{E}[ \frac{1}{T} \Sigma_{t=1}^{T}\ell(g(\cdot; f_{t}^{\prime\prime}, q_t/q_0); \mathcal{P}_{t}^{test})]$$

and
$$Z=\frac{1}{T} \Sigma_{t=1}^{T}\ell(g(\cdot; f_{0}, q_t/q_0); \mathcal{P}_{t}^{test})$$

According to [1],  Then, the Eq.(3) can be rewrote as
$$ X^{\mathcal{M}} - Y  \leq \mathcal{O}(\frac{K^{1/6}V_{T}^{1/3}}{\sigma^{2/3}T^{1/3}} + \frac{K}{\sigma \sqrt{T}})$$

If we define $\Delta = Z- Y$  as **the amount of improvement** ($\Delta \geq 0$ if we admit Eq. (5)).
Then, introducing feature update can make the original bound in Eq. (4) tighter by $\Delta$:
$$X^{\mathcal{M}} - Z \leq \mathcal{O}(\frac{K^{1/6}V_{T}^{1/3}}{\sigma^{2/3}T^{1/3}} + \frac{K}{\sigma \sqrt{T}}) - \Delta$$

The larger improvement we make by feature update (in other words larger $\Delta$), the tighter bound we can derive from the original version.
Thus, my question is: **could you please verify this point with empirical results to support your motivation in Eq. (5) in a quantitative manner?** Maybe we can fix an OLS method $\mathcal{M}$, and then choose different ways to obtain $f_{t}^{\prime\prime}$ (e.g., different $\ell_{ssl}$), then give a quantitative measure on the loss of $\mathcal{M}$-OFU (i.e., $X^{\mathcal{M}}$ defined above) with respect to the value of $\Delta$?

3. It seems the organization of the descriptions in Algorithm 1&2 is a little confusing and make the core steps of your methods less readable. For example, if we start Algorithm 1 from $t=1$, the step-1 of Algorithm 1 aims to return $f_{2}^{\prime} \leftarrow \textit{OLS-R}$. However, in the details of this step shown in Algorithm 2, it seems we need $f_{1}^{\prime\prime}$ returned by the step-3 of Algorithm of the previous loop. I failed to get what is the exact definition of $f_{1}^{\prime\prime}$.

References:

[1] Online label shift: Optimal dynamic regret meets practical algorithms. NeurIPS 2023.

---

> ### Author Response · Authors · 2023-11-16
> **Official Comment by Submission7196 Authors**
>
> Thank you for your constructive feedback!
>
> **Evidence between Equation (5) and results**: Thanks for raising this point! We agree that although our algorithm is theoretically motivated by Equation (5), stronger evidence between Equation (5) and the results can provide stronger motivation. To show this, we calculate the Pearson coefficient between the improvement from Equation (5) (RHS - LHS) and the improvement from OLS to OLS-OFU (i.e. loss$^{\rm OLS}-$loss$^{\rm OLS-OFU}$) across all experimental setups (dataset, online shift pattern, SSL method).  Pearson coefficient is between [-1, 1] and if it is positive, it indicates their positive correlation. The coefficients as evaluated are 0.70, 0.76, 0.54, 0.63, 0.58, and 0.75 when OLS is set as FTFWH, FTH, ROGD, ATLAS, UOGD, and FLHFTL respectively, which all show strong positive correlations. We will include this analysis in the revision.
>
> **Motivation of feature extraction refinement**: Please see our general reply “Clarifications on Contribution”.
>
> **Definition of $f_1''$**: Thank you for raising this clarification question. Both $f_1$ and $f_1''$ are defined as $f_0$ initially. We will address this in the revision.

---

> > ### Comment · Reviewer_VDvw · 2023-11-20
> > **Further comments after the authors' responses**
> >
> > I appreciate the effort from the authors. However, I think the authors did not fully get my question. What I want to verify is whether the amount of performance improvement is highly related to the amount of feature representation improvement under different SSL methods. That is because according to my above analysis, if the theoretical guarantee holds, it seems that a better SSL method (i.e., with larger RHS-LHS in Eq. (5)) will probably lead to more significant performance improvement (e.g., final accuracies) for each OLS method and each dataset. In other words, it would be better if the authors could compare the RHS-LHS in Eq. (5) as well as the performance improvement under different SSL methods for each OLS method and each dataset.
> >
> > I also read the reviews from the other reviewers and I noticed that some reviewers (i.e., xmwo and 5U9r) also have more or less doubt about the motivation for introducing the OFU process with the SSL method. I believe that this is the base of the theoretical guarantee of this paper. Without this, it seems that the gap between the theoretical guarantee of introducing the OFU process and the empirical improvement cannot be fully bridged.

---

### Official Review · Reviewer_TVnp · 2023-11-01

**Soundness:** 4 excellent
**Presentation:** 4 excellent
**Contribution:** 3 good
**Rating:** 8
**Confidence:** 4

**Summary:**

This paper introduces a novel method called OLS-OFU for improving online generalized label shift adaptation. The proposed method builds upon previous OLS methods by additionally leveraging self-supervised learning to improve feature representations and enhance predictive models. The paper provides theoretical analyses and empirical tests to demonstrate the effectiveness and robustness of OLS-OFU, especially in cases of domain shifts. The empirical tests were conducted on CIFAR-10 and CIFAR-10C datasets.

**Strengths:**

1. The paper is well written.
2. This paper addresses a seldom encountered yet highly realistic scenario called online generalized label shift. The proposed method builds upon previous OLS methods by additionally leveraging self-supervised learning to improve feature representations and enhance predictive models, offering a promising approach for addressing this problem.
3. This paper has conducted thorough experiments on various OLS methods and has also derived theoretical bounds separately for each method. The experimental results validate the theoretical analysis.

**Weaknesses:**

1. The proposed method doesn't introduce many novel ideas; it mainly involves modifying previous OLS methods under the setting of online generalized label shift and iteratively optimizing the model by incorporating additional self-supervised loss.
2. The experimental results are presented in the form of graphs and charts. It might be beneficial to include some quantitative tables for better clarity and to facilitate comparisons by others. Besides, the experimental dataset is relatively small. Conducting experiments on a larger dataset, such as ImageNet-C, would be more convincing.

**Questions:**

1. Can the online generalized label shift scenario be simplified as a combination of domain shift and label shift?
2. The selected self-supervised learning methods often require an additional branch, and their computational cost is not insignificant. Have you explored alternative lightweight methods for implementation, such as BN adaptation or entropy minimization?

---

> ### Author Response · Authors · 2023-11-16
> **Official Comment by Submission7196 Authors**
>
> Thank you for your positive feedback!
>
> **Novelty and contribution**: Please see our general reply “Clarifications on Contribution” about the novelty and technical contribution.
>
> **Quantitative Results**: Thank you for the suggestions. We will include tables in the appendix for completeness in the revision.
>
> **Experiments on the Imagenet and Imagenet-C**: Current OLS algorithms might not work very well when the number of classes is large (e.g. 1000 in Imagenet), because it is hard to estimate the confusion matrix (e.g. 1000 by 1000 matrix on Imagenet) but the confusion matrix estimation is a very crucial step in all OLS algorithms. We will include this limitation discussion in the revision.
>
> **Generalized label shift**: Yes, the definition of online generalized label shift implies it is a combination of domain shift and label shift, while domain shift is specified in this form: the two distributions conditioned on any label class can be aligned by the same mapping function $h$.
>
> **Lightweight SSL methods**: Besides MoCo, we also experimented with other two lightweight SSL methods: entropy minimization and rotation degree prediction. From Figure 2, 6 and 7, we can observe that both entropy minimization and rotation degree prediction provides very consistent improvement and MoCo provides the most significant improvement overall among all three SSL methods.

---

### Author Response · Authors · 2023-11-16
**Clarifications on Contribution**

We thank all reviewers for their insightful comments. We would like to clarify the contributions in our paper and will include these discussions in the revision.
1. Contribution in terms of performance improvement for online (generalized) label shift problem. All OLS algorithms in the literature don’t draw attention to the feature improvements, which is important in deep learning. Hence, we propose OLS-OFU to improve the feature extractor by SSL during the online stage. The improvement from OLS-OFU is at least as significant as the improvement in the OLS literature in online (generalized) label shifts. To see this, we compare
    - the improvement from FTH (proposed in the first OLS paper; [1]) to FLHFTL(SOTA OLS algorithm in the literature; [2]);
    - the improvement from FLHFTL to the FLHFTL-OFU.

    We attached the results in Figure 13 (Appendix D.9) in the current revision. The results show that the improvement from OLS-OFU is at least as significant as the improvement in the OLS literature. Moreover, the improvement from OLS to OLS-OFU is more consistent across different data settings. This consistency demonstrates the potential that OLS-OFU can have improvement for any future OLS algorithm.

2. Technical contribution. OLS-OFU provides a general algorithm to apply any SSL method to all OLS in the literature. This algorithm has some delicate designs to incorporate the feature updating module, which is *necessary* to guarantee the performance theoretically and empirically.
    - The design of the order between OLS-R and updating the feature extractor in Algorithm 1. It is necessary to run OLS-R before updating the feature extractor because ROGD, UOGD, and ATLAS require that the model being updated and the current label distribution estimator $s_t$ are independent of each other. If we first update the feature extractor in the model by $S_t$ in Algorithm 1, $s_t$ and the model in OLS-R would be correlated at the randomness of $S_t$.
    - The design of $f_t’’$. For those OLS (FLHFTL, FTH) that reweights some classifier,  they require the knowledge of both a classifier and the exact label marginal distribution where this classifier is trained[1, 2]. Therefore, in OLS-OFU, $f_{t}$ is not simply a reweight of $f_{t-1}$ because the label marginal distribution at time $t-1$ is not known. Instead, we design $f_{t}$ as a reweight of $f_t’’$, which is always a model for $q_0$ but with the latest feature extractor.

    We will add these details in the revision to better motivate the algorithm design.

[1] Online adaptation to label distribution shift. NeurIPS 2021.

[2] Online label shift: Optimal dynamic regret meets practical algorithms. NeurIPS 2023.

---

### Meta-Review · Area_Chair_macR · 2023-12-03

**Metareview:**

**Summary:**

This paper investigates the problem of online (generalized) label shift problem, and propose a method, namely OLS-OFU, by combining self-supervised learning (SSL) with existing online label shift (OLS) algorithms. The main role of SSL is to enhance the feature extraction by leveraging the unlabeled data. The authors have provided some experiments for verifying the effectiveness of their method.

**Strengths:**
1) The motivation for addressing the label shift problem is clear, especially in practical scenarios with streaming data. Moreover, this paper is well-written and easy to follow.
2) This paper further considers the online generalized label shift problem, which is more general than the setting studied by existing work.

**Weaknesses:**
1) The proposed method is a direct combination of existing techniques without incurring significant challenges.
2) The theoretical contribution of this paper is limited, because the theoretical results can be simply derived by following existing studies on OLS.
3) Current experimental results are not enough to verify the theoretical results and effectiveness of the proposed method.

**Justification For Why Not Higher Score:**

As summarized above, the reviewers still have some concerns about the novelty of the proposed method, the significance of the theoretical contribution, and the adequacy of experiments, which are not addressed by the authors' responses. After reading this paper myself, I also share the same concerns with these reviewers. I believe that these concerns are serious, and make this paper below the bar of ICLR.

**Justification For Why Not Lower Score:**

N/A

---

### Decision · Program_Chairs · 2024-01-16

Reject